# MedQ-Bench: Evaluating and Exploring Medical Image Quality Assessment Abilities in MLLMs

## Abstract

Medical Image Quality Assessment (IQA) serves as the first-mile safety gate for clinical AI, yet existing approaches remain constrained by scalar, score-based metrics and fail to reflect the descriptive, human-like reasoning process central to expert evaluation. To address this gap, we introduce **MedQ-Bench**, a comprehensive benchmark that establishes a **perception–reasoning paradigm** for language-based evaluation of medical image quality with Multi-modal Large Language Models (MLLMs). **MedQ-Bench** defines two complementary tasks: (1) **MedQ-Perception**, which probes low-level perceptual capability via human-curated questions on fundamental visual attributes; and (2) **MedQ-Reasoning**, encompassing both *no-reference* and *comparison reasoning* tasks, aligning model evaluation with human-like reasoning on image quality. The benchmark spans *5 imaging modalities* and *over 40 quality attributes*, totaling *2,600 perceptual queries* and *708 reasoning assessments*, covering diverse image sources including authentic clinical acquisitions, images with simulated degradations via physics-based reconstructions, and AI-generated images. To evaluate reasoning ability, we propose a *multi-dimensional judging protocol* that assesses model outputs along four complementary axes. We further conduct rigorous *human–AI alignment validation* by comparing LLM-based judgement with radiologists. Our evaluation of *14 state-of-the-art MLLMs* demonstrates that models exhibit preliminary but unstable perceptual and reasoning skills, with insufficient accuracy for reliable clinical use. These findings highlight the need for targeted optimization of MLLMs in medical IQA. We hope that MedQ-Bench will catalyze further exploration and unlock the untapped potential of MLLMs for medical image quality evaluation.

## 1 Introduction

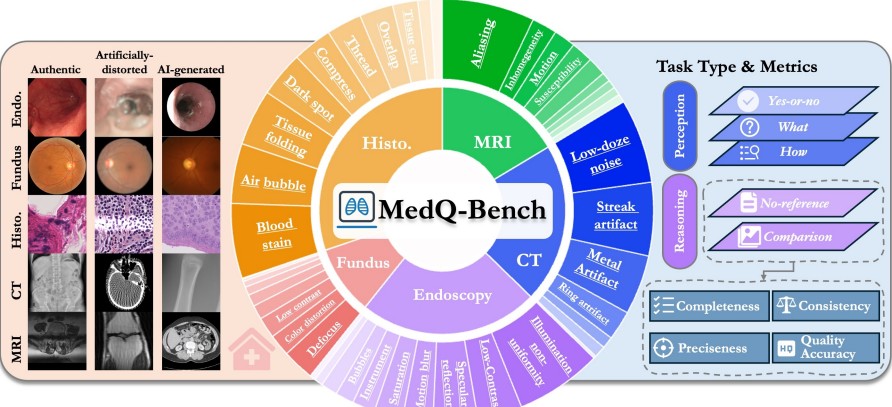

Figure 1: **MedQ-Bench** overview, evaluating MLLMs' abilities in medical image quality assessment with: (1) Comprehensive coverage: 3,308 samples across 5 modalities with 40+ degradation types. (2) Multi-faceted evaluation: perception-reasoning paradigm.

Medical Image Quality Assessment (IQA) determines whether imaging data can be reliably used for subsequent diagnostic interpretation and clinical decision-making (Lamard et al., 2024). In clinical

practice, multiple visual quality attributes of medical images directly influence diagnostic accuracy and patient safety (Rajpurkar et al., 2024), including sharpness, contrast adequacy, noise characteristics, artifact severity, etc. When these quality attributes are compromised, the resulting suboptimal images can lead to diagnostic errors, missed pathologies, or erroneous clinical interpretations, potentially causing severe patient harm and undermining the integrity of clinical decision-making processes (Blackmore et al., 2011).

Current medical IQA approaches predominantly produce scalar scores using (1) no-reference methods (Xun et al., 2025; Herath et al., 2025), which infer perceptual quality of an image through statistical feature extraction without a reference, and (2) full-reference similarity metrics such as PSNR, SSIM (Wang et al., 2004), and LPIPS (Zhang et al., 2018). These methods provide standardized evaluation metrics and enable automated IQA. However, they exhibit the following fundamental deficiencies. (1) *Poor generalization* (Herath et al., 2025). Medical image quality is influenced by complex and heterogeneous factors, including modality-specific degradations across MRI, CT, fundus photography, histopathology, and endoscopy. Yet, existing methods typically rely on simple regression models (Su et al., 2023) or handcrafted statistical indices (Dohmen et al., 2024), which generalize poorly to unseen distortions, new modalities, or different imaging protocols. (2) *Lack of human-like reasoning process for result interpretation*. Most methods produce scalar IQA scores, which do not fully reflect the causes of image quality degradation and may be unreasonable in certain cases. As illustrated in Figure 2, clinicians identify specific degradations and assess their clinical impact before making quality judgments. Traditional score-based metrics may contradict human judgment by favoring superficially smoother images while ignoring clinically important features. Such reasoning requires understanding the clinical significance of quality factors, which current paradigm approaches cannot effectively capture.

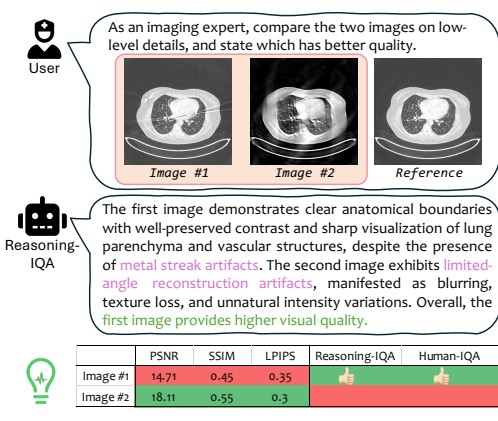

Figure 2: Comparison of Reasoning-IQA with score-based IQA paradigm. Unlike purely numerical scores, Reasoning IQA identifies distortion types and their relative impact, yielding results more consistent with Human-IQA.

Recent advances in multimodal large language models (MLLMs) have shown promising capabilities in medical visual reasoning tasks (OpenAI, 2023; Liu et al., 2024; Dai et al., 2024; Saab et al., 2024; Su et al., 2025). Theoretically, MLLMs could potentially address existing IQA challenges by decomposing quality assessment into interconnected subtasks: degradation identification, severity quantification, clinical impact analysis, and comparative reasoning. Unlike traditional approaches that yield opaque scores, MLLM-based assessment can provide explicit chains of thought (Wu et al., 2024a; You et al., 2024), offering interpretable and clinically meaningful evaluations. However, critical questions remain unanswered about MLLMs' actual capabilities in medical IQA: Can they truly generalize to the fine-grained, diverse, and complex quality factors spanning different imaging modalities? Do they possess genuine reasoning abilities to understand the clinical significance of various degradations? Existing MLLM evaluation frameworks focus mainly on natural images (Wu et al., 2024b) or high-level medical semantics (Ye et al., 2024), lacking systematic benchmarks that assess quality-related perceptual and reasoning skills across diverse medical modalities. This absence of specialized benchmarks has been a major barrier to developing and validating effective frameworks.

To bridge the gap between existing medical IQA methods, we propose a novel *perception–reasoning paradigm*. This paradigm mirrors clinicians' cognitive workflow: first perceiving quality-related attributes in images, assessing their severity, and evaluating their potential impact on clinical diagnosis, and then making overall quality judgments through logical reasoning. Building on this paradigm, we introduce **MedQ-Bench**, the first comprehensive benchmark that systematically evaluates the medical IQA capabilities of MLLMs. Our primary contributions are as follows:

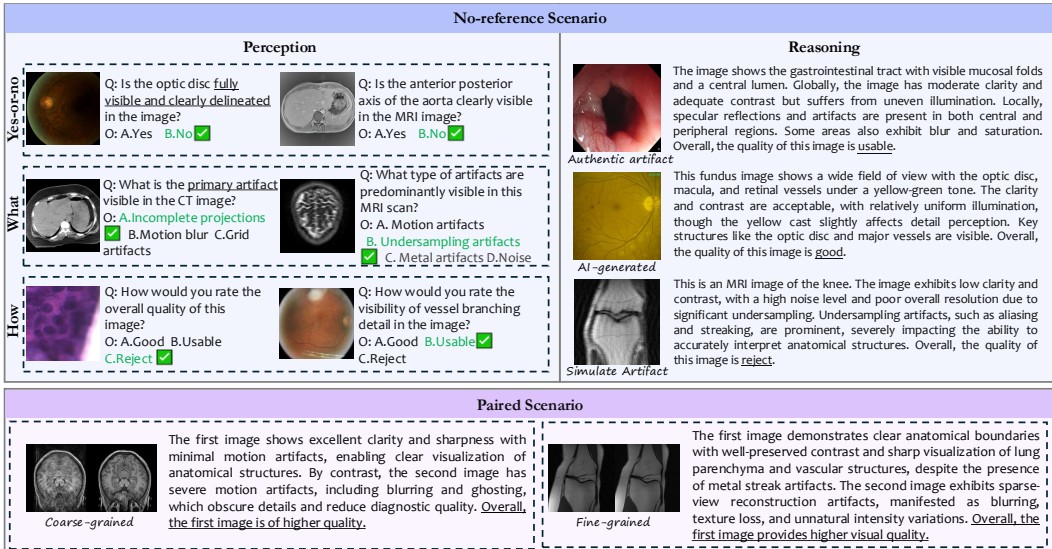

Figure 3: Examples of question types in MedQ-Bench, covering MCQA perception tasks (Yes-No / What / How), open-ended reasoning, and pair/multi-image comparison.

- **Pioneering evaluation framework for medical image quality assessment.** MedQ-Bench introduces a systematic evaluation methodology that comprehensively assesses both quality-based perceptual and reasoning capabilities for MLLMs. The framework extends beyond traditional IQA scoring to incorporate quality-related perception assessment, fine-grained comparative analysis, and quality-aware reasoning evaluation. The protocol supports both no-reference and full-reference paradigms, enabling systematic assessment ranging from coarse-grained to fine-grained perceptual discrimination tasks.

- **Multi-dimensional judging protocol with human–AI alignment validation.** To evaluate reasoning ability, we design a multi-dimensional judging protocol that scores model outputs along four complementary axes. We further perform rigorous human–AI alignment validation by comparing our LLM-based evaluations with radiologists, demonstrating the reliability of the proposed evaluation framework.

- **Comprehensive, clinically representative, multi-source dataset.** Covering *5 imaging modalities* and *40+ quality attributes*, **MedQ-Bench** blends authentic clinical images, simulated degraded images via physics-based reconstruction, and AI-generated images to encompass diverse real-world and controlled scenarios. This comprehensive dataset enables robust evaluation across both realistic clinical conditions and challenging scenarios.

- **Comprehensive empirical analysis.** We conduct extensive evaluations of state-of-the-art MLLMs, spanning open-source and commercial systems, both general-purpose and medical-specialized. Our systematic analysis reveals significant performance gaps in modality-specific perception capabilities, underscoring the need for targeted improvements for clinical readiness.

## 2 CONSTRUCTING THE MEDQ-BENCH

### 2.1 BENCHMARK SCOPE AND MODALITIES

Clinical image quality is fundamental to diagnostic reliability, yet existing evaluation methods rely primarily on score-based metrics that overlook the comprehensive assessment of image quality perception and reasoning capabilities. MedQ-Bench is specifically designed to systematically evaluate the visual quality perception and reasoning capabilities of multimodal large language models (MLLMs) within the medical imaging domain. Let $\mathcal{M} = \{M_1, M_2, \ldots, M_5\}$ represent the set of five medical imaging modalities, where each modality $M_i$ is associated with a distinct set of quality attributes $\mathcal{A}_i = \{a_{i,1}, a_{i,2}, \ldots, a_{i,k}\}$. The quality assessment task can be formulated as learning a

mapping function $f : \mathcal{I} \times \mathcal{Q} \rightarrow \mathcal{R}$ that takes an image $I \in \mathcal{I}$ and question $q \in \mathcal{Q}$ as input and produces a response $r \in \mathcal{R}$.

To capture the diversity and complexity of real-world clinical imaging, MedQ-Bench encompasses five representative modalities: Magnetic Resonance Imaging (MRI), Computed Tomography (CT), endoscopy, histopathology imaging, and fundus photography. Let $\mathcal{D}_i = \{d_{i,1}, d_{i,2}, \ldots, d_{i,n}\}$ denote the set of degradation types specific to modality $M_i$. Each modality exhibits distinct degradation characteristics due to its physical acquisition principles, where degradations can be modeled as transformations $T_d : \mathcal{I} \rightarrow \mathcal{I}'$ that modify the original image $I$ based on degradation type $d \in \mathcal{D}_i$. For instance, MRI is particularly susceptible to motion and magnetic susceptibility artifacts, and CT is prone to low-dose noise and metal-induced streak artifacts. This multi-modality design ensures that the benchmark reflects the broad spectrum of perceptual challenges encountered in practice.

For each modality, MedQ-Bench incorporates images from three complementary sources: authentic clinical images containing naturally occurring artifacts; synthetically degraded images that replicate modality-specific distortions in a controlled manner; and AI-generated or reconstructed images produced by enhancement, translation, or reconstruction models, which may introduce hallucinations or subtle structural inconsistencies. Let $\mathcal{S} = \{\mathcal{S}_{\text{real}}, \mathcal{S}_{\text{synth}}, \mathcal{S}_{\text{AI}}\}$ denote the three image sources, where each source $\mathcal{S}_k$ contributes a subset of images with specific degradation characteristics $\mathcal{D}_k \subseteq \bigcup_i \mathcal{D}_i$. This tri-source strategy enables the benchmark to cover both naturally occurring degradations and algorithm-induced artifacts, ensuring a balanced evaluation of MLLM robustness across real-world and algorithmic distortion scenarios.

## 2.2 BENCHMARK ON IQA PERCEPTION ABILITY

Before evaluating sophisticated reasoning capabilities, it is essential to establish whether MLLMs possess fundamental perceptual abilities to recognize basic image quality attributes.

### 2.2.1 QUESTION TYPES

The perception-focused MCQA setting evaluates direct visual perception using single-image prompts, without requiring domain-specific diagnostic reasoning. These tasks represent the most basic level of quality assessment capability, asking models to simply identify "what they see" rather than explain "why they see it." For each image, three canonical subtypes of questions are included: *(1) Yes-or-No*: Binary classification tasks where $\mathcal{R}_{\text{YN}} = \{0, 1\}$ and the model predicts $\hat{y} = \arg\max_{y \in \{0,1\}} P(y \mid I, q)$. Examples include "Is this image clear?" or "Does this image contain artifacts?" *(2) What*: Multi-class identification tasks where $\mathcal{R}_{\text{What}} = \{c_1, c_2, \ldots, c_K\}$ represents $K$ possible degradation types, and the model selects $\hat{c} = \arg\max_{c \in \mathcal{R}_{\text{What}}} P(c \mid I, q)$. These tasks ask models to identify specific types of artifacts or degradations present in the image. *(3) How*: Severity assessment tasks where $\mathcal{R}_{\text{How}} = \{s_1, s_2, \ldots, s_L\}$ represents $L$ severity levels, and the model predicts $\hat{s} = \arg\max_{s \in \mathcal{R}_{\text{How}}} P(s \mid I, q)$. These tasks evaluate the model's ability to assess the degree or intensity of observed quality issues.

### 2.2.2 QUADRANTS FOR LOW-LEVEL VISUAL CONCERNS

**Axis 1: No Degradation vs Degradation Severity Levels.** The primary axis differentiates medical images based on their quality degradation status: 1) *No Degradation* refers to medical images that maintain optimal quality standards without artifacts or distortions, and 2) degradation with Severity Levels encompasses images with varying degrees of quality issues, further subdivided into *mild Degradation* and *severe Degradation*.

**Axis 2: General Medical Questions vs Modality-specific Questions.** Quality perception in medical imaging intertwines with modality-specific technical characteristics. For instance, motion artifacts manifest differently in MRI versus CT scans. We curate *modality-specific questions* that require understanding unique technical characteristics of specific imaging modalities (e.g., "Does this MRI show susceptibility artifacts?"), while *general medical questions* focus on universal quality concepts applicable across modalities (e.g., "Is this image clear?"). This distinction evaluates both fundamental quality perception and specialized modality knowledge.

### 2.3 BENCHMARK ON IQA REASONING ABILITY

#### 2.3.1 NO-REFERENCE REASONING TASKS

While MCQA constrains answers to predefined choices, reasoning tasks assess a model's ability to autonomously describe and explain quality-related observations in natural language. These tasks require generating comprehensive responses $w_{1:T} = \{w_1, w_2, \ldots, w_T\}$ that systematically detail multiple aspects of image quality assessment: *(1)* modality and anatomical region identification; *(2)* specific quality degradation characterization including type and severity; *(3)* technical attribution of underlying causes; *(4)* assessment of diagnostic impact and clinical implications; and *(5)* definitive quality judgment with good/usable/reject recommendation. The reasoning tasks evaluate whether models can perform structured quality analysis that mirrors expert clinical assessment, moving beyond simple classification to demonstrate understanding of the relationship between technical image properties, degradation mechanisms, and clinical utility.

#### 2.3.2 COMPARISON REASONING TASKS

Many clinical workflows require comparative quality assessment between two versions of the same study, such as "original vs. reconstructed" or outputs from competing reconstruction algorithms. For image pairs $(I_A, I_B)$, the comparative task seeks to determine preference $P(I_A \succ I_B)$ based on overall quality assessment. Models must identify which image exhibits higher diagnostic quality and provide detailed explanations for their judgment, such as explaining why one reconstruction algorithm preserves anatomical detail better than another.

Comparative tasks are further categorized by the perceptual gap between images. *1) Coarse-grained* comparisons involve clearly visible quality differences, making them relatively straightforward for both humans and models. *2) Fine-grained* comparisons involve subtle differences in noise patterns, contrast, or structure fidelity, requiring heightened sensitivity to nuanced quality cues that may only be apparent upon careful inspection. This design enables separate evaluation of basic discrimination ability and advanced perceptual subtlety that approaches expert-level assessment sensitivity.

#### 2.3.3 EVALUATION METRICS

**Multi-dimensional judging protocol** The reasoning tasks require more nuanced evaluation approaches due to their subjective nature and the complexity of natural language responses. Recent studies have demonstrated GPT-4o to be a reliable evaluation tool for complex reasoning tasks. We assess model outputs $\mathcal{O}$ across four complementary dimensions, each scored on a discrete scale $s \in \{0, 1, 2\}$: **(1) Completeness.** $C(\mathcal{O}, \mathcal{R}) = \frac{1}{|\mathcal{K}_\mathcal{R}|} \sum_{k \in \mathcal{K}_\mathcal{R}} \mathbb{I}[k \in \mathcal{K}_\mathcal{O}]$ measures the coverage of key visual information from the reference description $\mathcal{R}$, where $\mathcal{K}_\mathcal{R}$ and $\mathcal{K}_\mathcal{O}$ represent the sets of key visual information in reference and output respectively. Higher scores indicate more comprehensive description of observable quality issues. **(2) Preciseness.** $P(\mathcal{O}, \mathcal{R}) = 1 - \frac{1}{|\mathcal{K}_\mathcal{O}|} \sum_{k \in \mathcal{K}_\mathcal{O}} \mathbb{I}[\text{contradict}(k, \mathcal{R})]$ quantifies consistency between model output and reference by penalizing semantic contradictions. **(3) Consistency.** $S(\mathcal{O}, \mathcal{R}) = f_{\text{consistency}}(\text{reasoning}(\mathcal{O}), \text{conclusion}(\mathcal{O}), \mathcal{R})$ evaluates the internal logical consistency between the reasoning path $\text{reasoning}(\mathcal{O})$ and the final quality judgment $\text{conclusion}(\mathcal{O})$, where $f_{\text{consistency}}$ returns a score based on logical coherence assessment. **(4) Quality Accuracy.** $Q(\mathcal{O}, \mathcal{R}) = \mathbb{I}[\text{comparison}(\mathcal{O}) = \text{comparison}(\mathcal{R})]$ assesses whether the final quality comparison judgment correctly identifies which image has higher quality, matching the reference assessment. This binary metric focuses on the correctness of the ultimate quality decision.

**Human–AI Alignment Validation** To ensure the reliability and validity of our automated evaluation, we conducted a rigorous alignment validation between GPT-4o judgments and expert assessments. A total of 200 cases were randomly sampled from the development dataset and independently evaluated by three board-certified medical imaging specialists under a double-blinded protocol.

For human–AI alignment, we employed quadratic weighted Cohen's kappa (Cohen, 1968) for ordinal ratings:

$$\kappa_w = 1 - \frac{\sum_{i,j} w_{ij} O_{ij}}{\sum_{i,j} w_{ij} E_{ij}}, \tag{1}$$

where $O_{ij}$ is the observed agreement matrix, $E_{ij}$ the expected agreement matrix, and $w_{ij} = \frac{(i-j)^2}{(k-1)^2}$ the quadratic weights penalizing larger disagreements more severely. We further conducted iterative prompt refinement to maximize concordance between GPT-4o and expert consensus. Final alignment results are reported in Section 3.4.

## 2.4 DATASET CONSTRUCTION AND ANNOTATION

Medical imaging experts designed structured seed templates defining the semantic intent and question structure for each question type. These seeds were expanded using GPT-4o as a controlled template instantiation tool under strict semantic constraints, with each generated question reviewed by three independent experts to eliminate model-specific linguistic biases. All question-answer pairs underwent multi-round expert validation: *(1)* independent annotation by three specialists following standardized protocols and modality-specific checklists; *(2)* cross-validation sessions where annotators identified divergent cases and conducted structured discussions; and *(3)* consensus resolution with majority voting for remaining conflicts. Complete details are provided in Appendix A.1.

## 3 RESULTS

To investigate MLLMs' image quality perception ability, we present a comprehensive evaluation of MedQ-Bench across 14 up-to-date popular MLLMs under zero-shot settings. We evaluate these 14 multimodal large language models across three categories: open-source MLLMs (Qwen2.5-VL-Instruct variants (Wang et al., 2024a), InternVL3 models (Chen et al., 2024b)), medical-specialized MLLMs (BiMediX2 (Peng et al., 2024), Lingshu (Wang et al., 2024b), MedGemma (Saab et al., 2024)), and commercial systems (GPT-5 (OpenAI, 2024b), GPT-4o (OpenAI, 2024a), Gemini-2.5-Pro (Reid et al., 2024), Grok-4 (xAI Team, 2024), Claude-4-Sonnet (Anthropic, 2024), Mistral-Medium-3 (Jiang et al., 2023)).

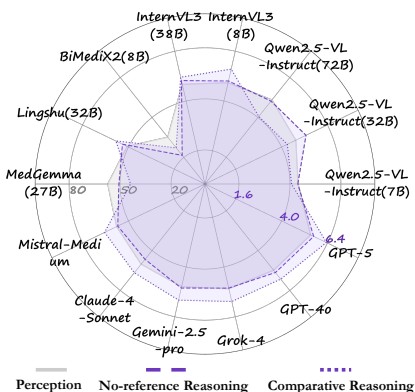

Figure 4: Overall Performance Results

## 3.1 FINDINGS ON PERCEPTION

To ensure rigorous and unbiased evaluation, the **MedQ-Perception** is equally divided into `dev` (Table 6, for prompt refinement) and `test` (Table 1, for final evaluation) subsets.

**Conclusion 1. Clear performance hierarchy emerges across model categories:** Our analysis reveals that most MLLMs perform above random guessing across all sub-tasks, indicating promising potential for domain generalization. The results demonstrate a clear performance hierarchy: closed-source frontier models achieve the highest scores, with GPT-5 leading at 68.97% on the test set. Among open-source models, Qwen2.5-VL-Instruct (72B) achieves the best performance at 63.14%, outperforming most commercial models, while *the best medical-specialized models underperform expectations*, with MedGemma (27B) achieving only 57.16%. More details are in Appendix A.5.1.

**Insufficiency 1. Substantial human-AI performance gap remains:** Another key finding emerges from our comparison with human performance, where we include both **human experts** (medical imaging technicians and medical imaging PhDs) and **non-experts** as reference points. The best AI model (GPT-5) significantly underperforms human experts (68.97% vs. 82.50%, a gap of 13.53%), yet outperforms non-experts by 6.47%. Given that these models have not undergone specialized training for medical image quality assessment, this suggests substantial potential for improvement in these MLLMs through further fine-tuning.

| Sub-categories | Perception | | | | Reasoning | | | | |
|---|---|---|---|---|---|---|---|---|---|
| Model (variant) | Yes-or-No↑ | What↑ | How↑ | Overall↑ | Comp.↑ | Prec.↑ | Cons.↑ | Qual.↑ | Overall↑ |
| *random guess* | 50.00% | 28.48% | 33.30% | 37.94% | | | | | |
| *Non-experts* | 67.50% | 57.50% | 57.50% | 62.50% | - | - | - | - | - |
| *Human experts* | 88.50% | 77.50% | 77.50% | 82.50% | - | - | - | - | - |
| Qwen2.5-VL-Instruct (7B) | 57.89% | 48.45% | 54.40% | 54.71%* | 0.715 | 0.670 | 1.855 | 1.127 | 4.367 |
| Qwen2.5-VL-Instruct (32B) | 67.38% | 43.02% | 58.69% | 59.31%* | 1.077 | 0.928 | **1.977** | 1.290 | 5.272 |
| InternVL3 (8B) | 72.04% | 47.67% | 52.97% | 60.08%* | 0.928 | 0.878 | 1.858 | 1.317 | 4.983 |
| InternVL3 (38B) | 69.71% | 57.36% | 52.97% | 61.00%* | 0.964 | 0.824 | 1.860 | 1.317 | 4.965 |
| Qwen2.5-VL-Instruct (72B) | 78.67% | 42.25% | 56.44% | 63.14%* | 0.905 | 0.860 | 1.896 | 1.321 | 4.982 |
| BiMediX2 (8B) | 44.98% | 27.52% | 27.81% | 35.10%* | 0.376 | 0.394 | 0.281 | 0.670 | 1.721 |
| Lingshu (32B) | 50.36% | 50.39% | 51.74% | 50.88%* | 0.624 | 0.697 | 1.932 | 1.059 | 4.312 |
| MedGemma (27B) | 67.03% | 48.06% | 50.72% | 57.16%* | 0.742 | 0.471 | 1.579 | 1.262 | 4.054 |
| Mistral-Medium-3 | 65.95% | 48.84% | 52.97% | 57.70%* | 0.923 | 0.729 | 1.566 | 1.339 | 4.557 |
| Claude-4-Sonnet | 71.51% | 46.51% | 54.60% | 60.23%* | 0.742 | 0.633 | 1.778 | 1.376 | 4.529 |
| Gemini-2.5-Pro | 75.13% | 55.02% | 50.54% | 61.88%* | 0.878 | 0.891 | 1.688 | **1.561** | 5.018 |
| Grok-4 | 73.30% | 48.84% | 59.10% | 63.14%* | 0.982 | 0.846 | 1.801 | 1.389 | 5.017 |
| GPT-4o | 78.48% | 49.64% | 57.32% | 64.79%* | 1.009 | 1.027 | 1.878 | 1.407 | 5.321 |
| GPT-5 | **82.26%** | **60.47%** | 58.28% | **68.97%** | **1.195** | **1.118** | 1.837 | 1.529 | **5.679** |

Table 1: Performance of different models on the MCQA perception and reasoning tasks. Significant differences in overall perception results compared with GPT-5 (paired t-test, $p < 0.05$) are indicated with asterisks.

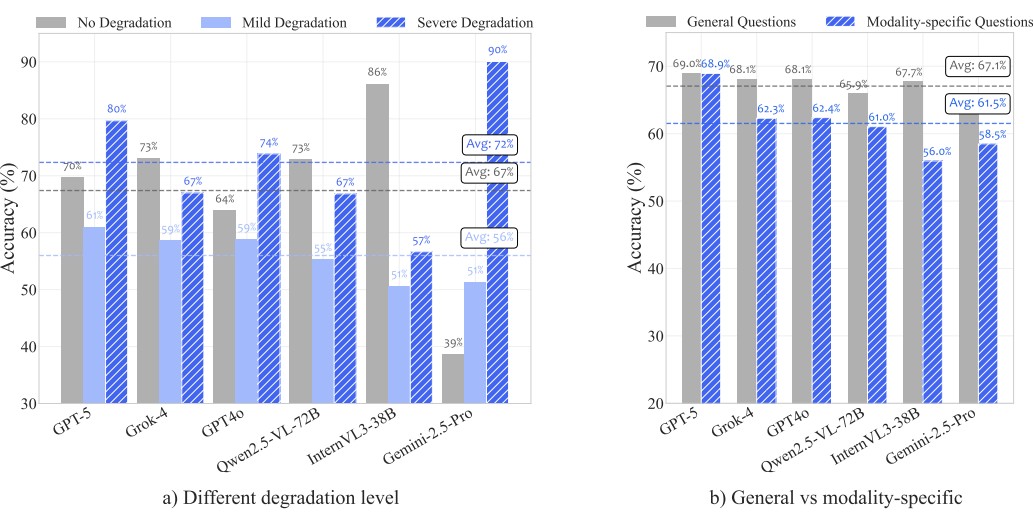

a) Different degradation level

b) General vs modality-specific

Figure 5: Performance analysis of MLLMs across different evaluation dimensions. (a) Different degradation level performance . (b) General vs modality-specific question.

**Insufficiency 2. The LVLMs are not robust among different perceptual types:** Task-specific analysis reveals distinct patterns across different evaluation dimensions. Performance analysis across different degradation levels (Figure 5(a)) demonstrates that mild degradation represents the most challenging detection scenario, with average accuracy dropping to 56% compared to 72% for no degradation and 67% for severe degradation. This indicates that subtle quality issues are harder to identify than obvious artifacts. Top-performing models like GPT-5 demonstrate a degree of consistency in performance across degradation levels. We further investigate the difference between general and modality-specific medical questions. As shown in Figure 5(b), most models perform better on general questions than on modality-specific tasks, whereas GPT-5 demonstrates the most balanced performance across question types. This suggests that robust medical image quality assessment requires specialized understanding of modality-specific visual features.

### 3.2 FINDINGS ON NO-REFERENCE REASONING

**Conclusion 2. Limited low-level visual reasoning capabilities across all models:** For no-reference reasoning capabilities (Table 1), GPT-5 still demonstrates the best performance, particu-

| Model | Comp.↑ | Prec.↑ | Cons.↑ | Qual.↑ | Overall↑ |
|---|---|---|---|---|---|
| Qwen2.5-VL-7B | 0.714 | 0.902 | 1.316 | 1.143 | 4.075 |
| Qwen2.5-VL-32B | 0.692 | 0.752 | 1.895 | 0.962 | 4.301 |
| Qwen2.5-VL-72B | 0.737 | 0.977 | 1.233 | 1.113 | 4.060 |
| InternVL3-8B | 0.985 | 1.278 | 1.797 | 1.474 | 5.534 |
| InternVL3-38B | 1.075 | 1.083 | 1.571 | 1.414 | 5.143 |
| BiMediX2-8B | 0.474 | 0.549 | 0.639 | 0.511 | 2.173 |
| MedGemma-27B | 0.684 | 0.692 | 1.128 | 1.000 | 3.504 |
| Lingshu-32B | 0.729 | 1.015 | 1.586 | 1.323 | 4.653 |
| Mistral-Medium-3 | 0.872 | 1.203 | 1.827 | 1.338 | 5.240 |
| Claude-4-Sonnet | 0.857 | 1.083 | 1.910 | 1.481 | 5.331 |
| Gemini-2.5-Pro | 1.053 | 1.233 | 1.774 | 1.534 | 5.594 |
| Grok-4 | 1.150 | 1.233 | 1.820 | 1.459 | 5.662 |
| GPT-4o | 1.105 | 1.414 | 1.632 | 1.562 | 5.713 |
| GPT-5 | 1.293 | 1.556 | 1.925 | 1.564 | 6.338 |

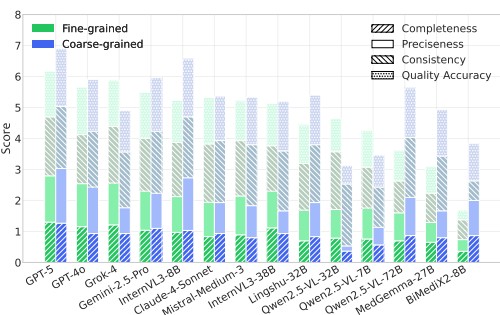

Figure 6: Comparative reasoning performance analysis. Left: Detailed performance scores across four evaluation dimensions for all models. Right: Visual comparison of overall performance patterns across model categories.

larly excelling in the relevance dimension. However, even the most advanced MLLMs fail to achieve excellent scores in completeness and preciseness, with the highest scores being only 1.293/2.0 for completeness and 1.556/2.0 for preciseness. In general, most models only reach an acceptable baseline level. Current MLLM models possess relatively limited and elementary low-level visual reasoning abilities, struggling to provide complete and accurate descriptions of low-level visual information. The consistently high consistency scores indicate that most MLLMs can follow abstract instructions reasonably well, suggesting that the main bottleneck for improving MLLM descriptive capabilities lies in the perception of low-level attributes rather than instruction following.

### 3.3 FINDINGS ON COMPARISON REASONING

**Insufficiency 3. Paired comparison reveals fundamental limitations in fine-grained analysis:** Paired image comparison tasks pose the greatest challenge to current multimodal large language models (MLLMs), requiring models to perform fine-grained quality comparisons between similar images that may only differ by varying degrees. We evaluate model performance across two difficulty levels: fine-grained differences and coarse-grained differences. Figure 6 (right) presents detailed performance analysis across different difficulty levels, with more complete tabular results available in Table 10 in the appendix. Overall, most models perform better under coarse-grained differences, while a few models, such as Grok-4 and Qwen2.5-VL-7B/32B, perform better under fine-grained differences but lose performance on coarse-grained tasks. Among them, GPT-5 achieved the highest overall score, while medical-specialized models such as BiMediX2 showed notably insufficient performance.

### 3.4 HUMAN–AI ALIGNMENT VALIDATION

**Strong human-AI alignment validates our evaluation framework:** To validate the reliability of our automated evaluation approach, we conducted a comprehensive human-AI alignment study comparing human expert assessments with GPT-4o automated scoring. We evaluated 200 randomly sampled image quality assessments across three key dimensions: completeness, preciseness, and consistency. The confusion matrices in the appendix (Figure 13) demonstrate strong alignment between human expert scores and GPT-4o automated evaluation across all three dimensions, with consistently high accuracy rates: 83.3% for completeness, 87.0% for preciseness, and 90.5% for consistency, with all individual class recall rates exceeding 80%.

These results validate that our automated quality assessment system achieves strong alignment with human expert judgment across all evaluation dimensions, with high accuracy rates demonstrating that our evaluation framework can serve as a reliable substitute for human evaluation. Beyond accuracy, we further assessed inter-rater agreement using quadratic weighted Cohen's $\kappa_w$ (Table 13), achieving consistently high values (0.774–0.985) that confirm substantial agreement beyond chance and validate our framework as a reliable surrogate for large-scale human evaluation.

## 4 RELATED WORK

**Medical Multimodal Large Language Models and Benchmarks.**   Multimodal Large Language Models (MLLMs) have demonstrated remarkable capabilities in understanding and reasoning about visual content through natural language. General-purpose models like GPT-4V (OpenAI, 2023), LLaVA (Liu et al., 2024), and Qwen-VL (Wang et al., 2024a) have shown strong performance across diverse vision-language tasks. To address healthcare-specific requirements, medical-specialized variants such as (Wang et al., 2024b; Saab et al., 2024; Peng et al., 2024; Su et al., 2025; Xu et al., 2025) have emerged through domain-targeted pretraining and alignment. Recent medical benchmarks have been developed to evaluate these models systematically, including (Ye et al., 2024), which provides comprehensive multimodal evaluation for general medical AI. However, existing medical benchmarks focus primarily on high-level diagnostic tasks rather than low-level perceptual quality assessment (Chen et al., 2024a).

**Score-based Image Quality Assessment.**   Traditional image quality assessment methods produce numerical scores to quantify image quality, categorized into No-Reference (NR), Full-Reference (FR), and Reduced-Reference approaches. NR methods like BRISQUE (Mittal et al., 2012), NIQE (Zhang et al., 2015), and deep learning approaches including CNNIQA (Kang et al., 2014) and MUSIQ (Ke et al., 2021) assess quality without reference images. FR methods compare against pristine references using metrics like PSNR, SSIM (Wang et al., 2004), VIF (Sheikh & Bovik, 2006), and learned perceptual metrics like LPIPS (Zhang et al., 2018). Recent advances include transformer-based approaches like TReS (Golestaneh et al., 2022) and quality-aware pretraining methods. However, these methods yield only scalar scores, offering limited interpretability regarding specific quality factors, and such technical measures often show weak alignment with clinical workflows (Zhang et al., 2024; Blackmore et al., 2011).

**MLLM-based Image Quality Assessment.**   Recent advances have introduced multimodal language models for image quality assessment (IQA), which enable more interpretable and reasoning-based evaluation. For example, Q-Instruct (Wu et al., 2024a) and DepictQA (You et al., 2024) generate natural language descriptions of quality factors, while Q-Bench (Wu et al., 2024b) offers a systematic framework for evaluating low-level vision tasks. Building on this line, IQAGPT (Chen et al., 2023) integrates vision-language models with ChatGPT for CT image quality assessment, showing the feasibility of producing both quality scores and textual reports. However, its scope is limited to CT images and remains focused on score prediction rather than comprehensive reasoning. Likewise, Ultrasound-QBench (Miao et al., 2025) provides evaluation for ultrasound imaging but restricts tasks to classification and scoring within a single modality.

## 5 CONCLUSION

We introduced **MedQ-Bench**, the first benchmark to systematically evaluate medical image quality assessment (IQA) capabilities of multimodal large language models through a perception–reasoning paradigm. Unlike conventional score-based metrics, **MedQ-Bench** jointly assesses quality-related perception and reasoning across five imaging modalities and more than forty degradation types via three complementary tracks: perception tasks, no-reference reasoning, and paired comparison reasoning. Our large-scale zero-shot evaluation of 14 state-of-the-art MLLMs, including open-source, medical-specialized, and commercial systems, yields several key findings. Substantial performance gaps remain between AI models and human experts, particularly in detecting subtle degradations critical to clinical practice. Current models exhibit preliminary but unstable perceptual and reasoning abilities, often failing to produce complete and precise quality descriptions. Medical-specialized models unexpectedly underperform general-purpose ones, calling into question the effectiveness of current domain adaptation strategies. Moreover, models show marked weaknesses in fine-grained comparisons and mild degradation detection, precisely where reliable quality control is most needed. By moving beyond high-level diagnostic reasoning toward foundational quality perceptual and reasoning skills, **MedQ-Bench** establishes a clinically grounded and interpretable standard for measuring and advancing medical IQA. We anticipate that it will inform the development of MLLMs with stronger low-level visual understanding and trustworthy reasoning, paving the way for safe and reliable integration of automated quality control into clinical imaging workflows.

REPRODUCIBILITY STATEMENT

To support reproducibility, the benchmark dataset and evaluation code are available at `https://anonymous.4open.science/r/MedQ-Bench`.

ETHICS STATEMENT

This work involves the collection and curation of medical images to construct the MedQ-Bench benchmark. All images were either obtained from publicly available, ethically released datasets or collected under institutional approval with strict de-identification to remove patient-identifying information. No protected health information (PHI) is included. All human annotations were performed by qualified medical imaging experts under anonymized, non-patient-identifiable conditions. The benchmark is intended solely for research and model evaluation; it is not intended to guide real clinical decision-making and should not replace expert judgment.

We have carefully considered potential risks of misuse, such as overreliance on automated medical image quality assessment in clinical workflows without expert oversight. To mitigate these risks, we release the benchmark strictly for research purposes with clear disclaimers that models evaluated with MedQ-Bench must undergo further clinical validation before deployment. No human subjects were directly involved in experiments beyond expert annotation, and no diagnostic or treatment decisions were made in the course of this work.

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

# A APPENDIX

APPENDIX TABLE OF CONTENTS

## A.1 DATA CONSTRUCTION PIPELINE AND QUALITY CONTROL

The construction of MedQ-Bench involved a systematic multi-stage pipeline for collecting, curating, and annotating medical images across five modalities. This section provides detailed information about our comprehensive data sourcing strategies, quality control measures, and annotation protocols, with particular emphasis on the diverse sources and label types that enable robust evaluation of low-level visual perception capabilities.

**Comprehensive Data Sources and Acquisition Strategy.** We employed a three-channel data collection strategy: public datasets + imaging department collaboration + synthetic generation. On one hand, we conducted comprehensive internet searches for 2D/3D medical quality-related datasets; on the other hand, we collaborated with hospitals to obtain ethically approved clinical data. From this massive data pool, we ultimately selected the datasets shown in Table 3, covering 5 medical imaging modalities to ensure universality and clinical relevance of data sources. For images, we adhere to the SA-Med2D-20M (Ye et al., 2023) protocol, transforming all 2D/3D medical images into 2D RGB images for further evaluation. Table 3 provides a complete overview of all datasets integrated into MedQ-Bench, including specific modalities, sample quantities, label types, and acquisition status. The table demonstrates the comprehensive scope of our data collection effort, spanning established clinical research datasets and custom synthetic degradation collections, and AI-generated images. All collected images were anonymized, with all patient-identifying information systematically removed using automated de-identification pipelines validated against clinical privacy requirements.

**Expert-designed Seed Perception Questions.** The construction process began with a panel of medical imaging specialists who designed seed questions covering diverse modalities, degradation types, and task formats. Each seed is a structured template with predefined placeholders (e.g., [artifact_type], [quality_attribute]) whose values are constrained by expert-defined taxonomies. These [QUESTION_SEEDS] are defined as:

```
Type 0 (Yes-No):
  T0.1: "Does this image show evidence of [artifact_type]?"
  T0.2: "Is the [artifact_type] fully visible?"

Type 1 (What):
  T1.1: "What is the primary [assessment_target]?"
  T1.2: "What [quality_dimension] issue is most prominent?"

Type 2 (How):
  T2.1: "How would you rate the severity of [artifact_type]?"
  T2.2: "How would you rate the overall [quality_dimension]?"
```

**Controlled Question Expansion.** To generate questions for each sample, we employed GPT-4o as a controlled question generator. GPT-4o served solely as a template instantiation tool under strict constraints, receiving explicit system prompts:

```
System: You are a [modality] image quality assessment expert.
Task: As an [modality] image quality assessment expert, please
questions based on the following information:

Given:
- Image path: [IMAGE_PATH]
- Artifacts of this image: [ARTIFACT]
- Modality-specific assessment criteria: [ARTIFACT_LIST],
[TECHNICAL_QUALITY_ASSESSMENT_CRITERIA], and other information.

Your task is to generate EXACTLY 3 questions that assess the
image quality based on the criteria and the provided image.
Consider the following question seeds and constrains:
```

```
Question seeds:
[QUESTION_SEEDS]

Constraints:
1. Preserve each seed's semantic intent and structure exactly
2. Substitute [artifact_type] with detected artifacts
3. Generate one Yes-or-No, one What, and one How question, with
each question focusing on one of four quality concerns:
0:technical distortion, 1:global visual quality, 2:local structure
visibility, 3:lighting/brightness/color/others
4. Only focus on low-level attributes.
```

This ensures expansion is limited to placeholder substitution and minimal paraphrasing. Every expanded question underwent review and revision by medical imaging experts to eliminate model-specific biases and ensure alignment with expert-designed seeds.

**Multi-round Expert Validation.** We manually annotated the answers for the generated questions to ensure correctness, consistency, and alignment with the intended low-level quality assessment labels. All question–answer pairs underwent rigorous multi-stage human annotation and verification using a structured annotation interface, as shown in Figure 7 and 8. The multi-round validation process consisted of three phases: *(1)* Independent annotation by three medical imaging specialists, each following the standardized perception and reasoning annotation protocols as well as modality-specific checklists to minimize arbitrary judgments; *(2)* Cross-validation sessions in which annotators reviewed each other's labels, identified divergent cases, and conducted structured discussions based on explicit visual and clinical evidence. For reasoning tasks in particular, discrepancies were further examined by aligning observation-level evidence, rechecking modality-specific rules, reconciling classification boundaries (e.g., "usable" versus "reject"), and harmonizing reasoning chains to ensure logical consistency between visual attributes, modality rules, and textual explanations; *(3)* Consensus resolution, in which labels and reasoning descriptions were revised once agreement was reached, and the validated evidence and reasoning sequence were consolidated into a single final reasoning trajectory. For the rare cases where conflicts persisted, whether in evidential interpretation or in final conclusions, the final label and reasoning statement were determined by majority voting. This unified protocol reduces idiosyncratic variability across annotators and establishes a stable and clinically grounded expert standard. Finally, the dataset was randomly partitioned into development and test sets of equal size.

**Reasoning Annotation Standards and Workflow.** For the MedQ-Reasoning tasks, we established specific annotation standards to ensure consistent and clinically relevant quality assessment descriptions. Expert annotators followed a structured reasoning workflow that emphasized systematic analysis and transparent decision-making processes. The reasoning annotation protocol involved a sequential four-step process: *(1)* Visual Analysis Phase: Systematic examination of perceptual attributes such as noise, blur, artifacts, contrast, and resolution, avoiding any high-level diagnostic interpretation; *(2)* Modality-Specific Assessment: Targeted evaluation of quality dimensions specific to each imaging modality (e.g., streak artifacts in CT, motion artifacts in MRI, staining uniformity in histopathology), following standardized checklists for each modality type; *(3)* Quality Classification: Application of a three-tier system based on accumulated evidence from steps 1-2: "good" (no significant quality issues affecting clinical utility), "usable" (minor quality issues that do not compromise diagnostic accuracy), and "reject" (severe quality degradation requiring repeat imaging); *(4)* Structured Description Generation: Creation of comprehensive yet concise descriptions (3-5 sentences) that logically connect the observed visual attributes to the final quality judgment, ensuring clear reasoning traceability from observation to conclusion. This step-by-step reasoning flow ensures that all quality assessments follow a consistent analytical framework, with each conclusion being explicitly grounded in observable visual evidence rather than subjective impressions.

**Dataset Composition and Balance.** Each modality contributes proportionally to maintain representational balance, and degradation types are systematically distributed to avoid bias toward any particular quality issue.

**Fine-Grained vs. Coarse-Grained Definition.**    The fine-grained versus coarse-grained distinction for comparison tasks was defined through a structured expert review process. Three board-certified radiologists independently reviewed all image pairs and categorized them based on how difficult the quality difference is to perceive: coarse-grained pairs exhibit clear, immediately noticeable degradations, whereas fine-grained pairs involve subtle differences that require deliberate inspection. After independent labeling, disagreements were resolved through discussion and consensus, ensuring a stable expert standard.

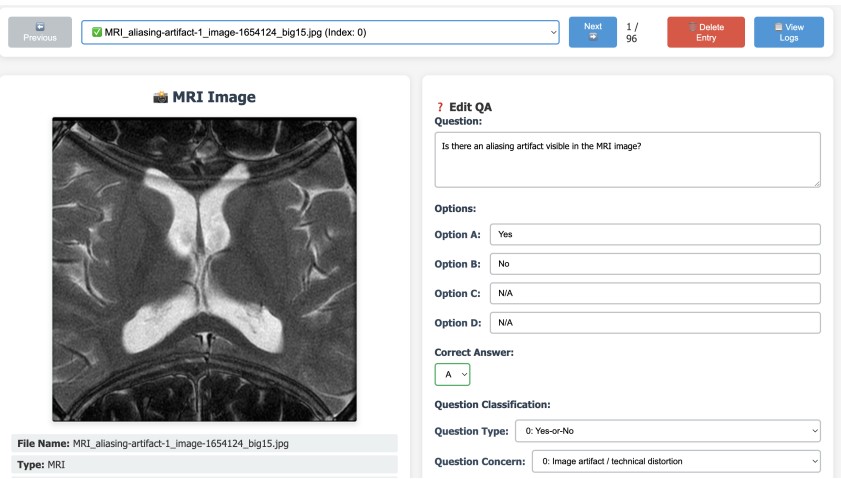

Figure 7: Interface for the MedQ-MCQA dataset.

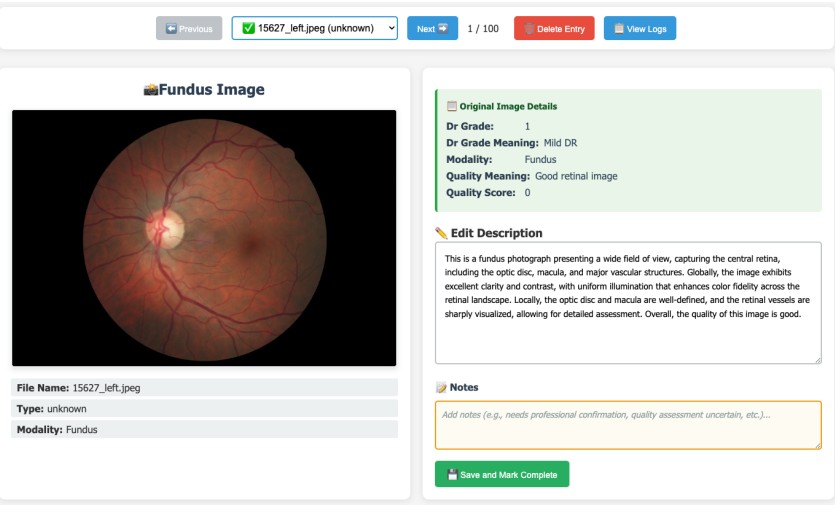

Figure 8: Interface for the MedQ-Reasoning dataset.

## A.2    DETAILED BENCHMARK STATISTICS

### A.2.1    DATASET COMPOSITION BY SOURCE AND MODALITY

**Dataset Composition.**    The MedQ-Bench dataset consists of 3,308 samples distributed across three primary source types (Table 2). The dataset covers five major medical imaging modalities with detailed breakdown by source type and specific datasets shown in Table 3.

| Source Type | Authentic | Simulate | AI-Generated |
|---|---|---|---|
| **Percentage** | 41.3% | 33.9% | 24.8% |

Table 2: Distribution of MedQ-Bench dataset by source type.

| Modality | Source Type | Dataset | Samples | % of Modality | Total Samples |
|---|---|---|---|---|---|
| CT | Authentic | Radiopaedia | 239 | 27.2% | |
| | Simulate | AAPM CT-MAR (AAPM CT-MAR Challenge Organizers, 2023) | 613 | 69.8% | |
| | AI-generated | AIsynthesis (subset) | 26 | 3.0% | 878 |
| MRI | Authentic | Radiopaedia | 130 | 15.0% | |
| | Authentic | MR-ART (Ádám Nárai et al., 2022) | 68 | 7.9% | |
| | Authentic | FSL Example MRI Artifacts | 43 | 5.0% | |
| | Simulate | FastMRI Zbontar et al. (2018) | 454 | 54.4% | 848 |
| | Simulate | 5T MRI Data | 55 | 6.4% | |
| | AI-generated | AIsynthesis | 98 | 11.3% | |
| Histopathology | Authentic | HistoArtifacts (Kanwal, 2024) | 220 | 29.0% | |
| | AI-generated | HARP (Fuchs et al., 2024) | 470 | 62.0% | 758 |
| | AI-generated | AIsynthesis | 68 | 9.0% | |
| Endoscopy | Authentic | EndoCV2020 (Polat et al., 2020) | 470 | 84.7% | 555 |
| | AI-generated | AIsynthesis | 85 | 15.3% | |
| Retinal | Authentic | EyeQ (Fu et al., 2019) | 197 | 73.2% | 269 |
| | AI-generated | AIsynthesis | 72 | 26.8% | |
| | | **Overall Total** | **3,308** | | |

Table 3: Comprehensive breakdown of dataset composition.

**Detailed Simulation Methods for Synthetic Degradations.** The simulated CT degradations in AAPM CT-MAR were reconstructed using several algorithms: SIRT, FBP (Kak & Slaney, 2001), and FISTA (Beck & Teboulle, 2009). Specifically, CT artifacts were systematically simulated to include three primary degradation types: *(1)* limited-angle artifacts, *(2)* metal artifact reduction, and *(3)* sparse-view artifacts. For MRI degradations, we primarily simulated acceleration artifacts and motion artifacts using established computational frameworks. Acceleration artifacts were generated using SigPy[1] and TorchIO[2], implementing both DDNM (Wang et al., 2022) and wavelet-based reconstruction methods (Guerquin-Kern et al., 2011). Additionally, our 5T MRI data were acquired from private clinical collections using the uMR Jupiter 5T system, obtained under institutional ethical approval with comprehensive patient anonymization protocols.

To generate synthetic images across diverse medical imaging modalities, we employed BAGEL fine-tuned on domain-specific medical datasets (Deng et al., 2025). This approach ensured that synthetic degradations maintained clinical realism while providing controlled quality variations essential for comprehensive benchmark evaluation.

### A.2.2 DISTRIBUTIONS OF TASK TYPES AND DEGRADATION LEVELS IN MEDQ-PERCEPTION

| Question Type | Percentage |
|---|---|
| Modality-specific | 57.2% |
| General | 42.8% |
| **Total** | **100.0%** |

Table 4: MedQ-Perception: Distribution of tasks by question type.

| Degradation Level | Percentage |
|---|---|
| No Degradation | 23.8% |
| Mild Degradation | 44.6% |
| Severe Degradation | 31.6% |
| **Total** | **100.0%** |

Table 5: MedQ-Perception: Distribution of degradation severity levels.

Table 4 and Table 5 present the distribution of MedQ-Perception tasks across question types and degradation severity levels.

### A.2.3 DISTRIBUTION OF LOW-LEVEL ATTRIBUTIONS

Figure 9 illustrates the comprehensive distribution of low-level quality attributes across five medical imaging modalities, covering over 40 distinct degradation types including modality-specific artifacts and general quality issues.

---

[1]https://sigpy.readthedocs.io/en/latest/

[2]https://github.com/TorchIO-project/torchio

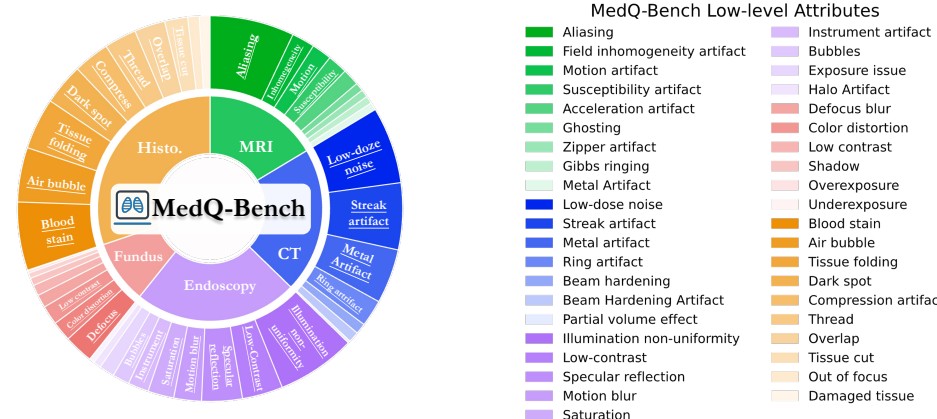

Figure 9: Distribution of low-level attributions across imaging modalities and distortion types in MedQ-Bench.

## A.3 EVALUATION PROMPT

---

**Single-Image Perception Task Prompts**

**Yes-No / What / How Question Template:**

```
You are an expert in medical image quality assessment.
Please carefully observe this medical image and answer
the following question:
```

---

**Reasoning Task Prompts**

**No-reference Reasoning Template:**

```
As a medical image quality assessment expert, provide
a concise description focusing on low-level appearance
of the image in details.  Conclude with ``Overall, the
quality of this image is [good/usable/reject]''.  Please
provide a comprehensive but concise assessment in 3-5
sentences.
```

**Comprehensive Reasoning Template:**

```
As a medical image quality assessment expert, provide
a concise description comparing two images focusing
on low-level appearance.  Conclude with which image
has higher quality.  Please provide comprehensive but
concise assessment in 3-5 sentences.
```

---

**Complete Evaluation Prompt Templates for No-reference Reasoning Tasks**

**Completeness Evaluation Prompt:**

```
#System: You are a helpful assistant.
#User: Evaluate whether the description [MLLM DESC] completely
includes the low-level visual information in the reference
description [GOLDEN DESC]. Please rate score 2 for completely
or almost completely including reference information, 0 for not
including at all, 1 for including part of the information or
similar description.
Please only provide the result in the following format: Score:
```

**Preciseness Evaluation Prompt:**

```
#System: You are a helpful assistant.
#User: The precision metric evaluates whether the low-level
description is consistent with the reference and reasonably
aligned with the final quality judgment. Minor wording
differences or small omissions that do not change the overall
meaning should still be considered consistent.
Only penalize clear contradictions with the reference, such as
describing blur for clear, noisy for clean, motion-free for
motion artifacts, noise-free for low-dose noise, etc.
Evaluate whether output [MLLM DESC] reasonably reflects
reference [GOLDEN DESC].
Please rate score 2 for overall consistency and no major
contradictions with the quality conclusion, 1 for partial
consistency or very few minor contradictions, and 0 for obvious
contradictions or misalignment with the quality conclusion.
Please only provide the result in the following format: Score:
```

**Consistency Evaluation Prompt:**

```
#System: You are a helpful assistant.
#User: Evaluate the internal consistency between the reasoning
path (description of image problems) and the final quality
judgment in [MLLM DESC]. The reasoning should logically support
the final quality conclusion. For example, if many serious
problems are described, the final quality should be "reject";
if minor problems are described, it should be "usable"; if no
or very few problems are described, it should be "good".
Compare with the reference [GOLDEN DESC] to understand the
expected reasoning-conclusion relationship.
Please rate score 2 for highly consistent reasoning and
conclusion, 1 for partially consistent with minor logical gaps,
and 0 for major inconsistency between described problems and
quality judgment.
Please only provide the result in the following format: Score:
```

**Quality Accuracy Evaluation Prompt:**

```
#System: You are a helpful assistant.
#User: Evaluate the accuracy of the final quality judgment in
[MLLM DESC] compared to the reference [GOLDEN DESC].
The quality levels have a progressive relationship: reject <
usable < good. Consider the distance between predicted and
reference quality:
Please rate score 2 for exactly matching the reference quality
level, 1 for adjacent level difference (e.g., usable vs good,
or reject vs usable), and 0 for distant level difference
(reject vs good) or completely incorrect quality assessment.
Please only provide the result in the following format: Score:
```

**Complete Evaluation Prompt Templates for Comparison Reasoning Tasks**

**Completeness Evaluation Prompt:**

```
#System: You are a helpful assistant.
#User: Evaluate whether the description [MLLM DESC] completely
includes the low-level visual information in the reference
description [GOLDEN DESC]. Please rate score 2 for completely
or almost completely including reference information, 0 for not
including at all, 1 for including part of the information or
similar description.
Please only provide the result in the following format: Score:
```

**Preciseness Evaluation Prompt:**

```
#System: You are a helpful assistant.
#User: The precision metric evaluates whether the low-level
description is consistent with the reference and reasonably
aligned with the final quality judgment. Minor wording
differences or small omissions that do not change the overall
meaning should still be considered consistent.
Only penalize clear contradictions with the reference, such as
describing blur for clear, noisy for clean, motion-free for
motion artifacts, noise-free for low-dose noise, etc.
Evaluate whether output [MLLM DESC] reasonably reflects
reference [GOLDEN DESC].
Please rate score 2 for overall consistency and no major
contradictions with the quality conclusion, 1 for partial
consistency or very few minor contradictions, and 0 for obvious
contradictions or misalignment with the quality conclusion.
Please only provide the result in the following format: Score:
```

**Consistency Evaluation Prompt:**

```
#System: You are a helpful assistant.
#User: Evaluate the internal consistency between the reasoning
path (comparative description of image problems) and the final
quality comparison judgment in [MLLM DESC]. The reasoning
should logically support the final comparison conclusion.
Compare with the reference [GOLDEN DESC] to understand the
expected reasoning-conclusion relationship for image comparison.
Please rate score 2 for highly consistent reasoning and
comparison conclusion, 1 for partially consistent with minor
logical gaps, and 0 for major inconsistency between described
comparative problems and quality comparison judgment.
Please only provide the result in the following format: Score:
```

**Quality Accuracy Evaluation Prompt:**

```
#System: You are a helpful assistant.
#User: Evaluate the accuracy of the final quality comparison
judgment in [MLLM DESC] compared to the reference [GOLDEN DESC].
The comparison should correctly identify which image has higher
quality based on the described visual characteristics.
Please rate score 2 for exactly matching the reference quality
comparison, and 0 for completely incorrect quality comparison
(opposite conclusion) or unreasonable assessment.
Please only provide the result in the following format: Score:
```

## A.4 COMPLETE EXPERIMENTAL RESULTS

### A.4.1 EXPERIMENTAL SETUP

In this study, we evaluated various large vision-language models (LVLMs), encompassing medical-specialized models, open-source models, and closed-source API general-purpose models. Model weights were obtained from their respective official Hugging Face repositories. The evaluation work was conducted using the VLMEvalKit framework[3].

The evaluation was performed under a "zero-shot" setting. Specifically, our evaluation prompts contained no example demonstrations, and models had to complete task reasoning without any related training or examples. This approach better tests the models' generalization capabilities and understanding abilities, examining their performance when faced with novel problems. All tests were executed on NVIDIA A100 GPUs with 80GB memory.

### A.4.2 DETAILED MODEL PERFORMANCE

| Sub-categories | Perception (Dev) | | | | Reasoning (Dev) | | | | |
|---|---|---|---|---|---|---|---|---|---|
| Model (variant) | Yes-or-No↑ | What↑ | How↑ | Overall↑ | Comp.↑ | Prec.↑ | Cons.↑ | Qual.↑ | Overall↑ |
| Qwen2.5-VL-Instruct (7B) | 62.71% | 45.26% | 53.93% | 56.32% | 0.688 | 0.615 | 1.869 | 1.122 | 4.294 |
| Qwen2.5-VL-Instruct (32B) | 64.43% | 44.40% | 56.20% | 57.78% | 1.036 | 0.896 | 1.959 | 1.253 | 5.144 |
| Qwen2.5-VL-Instruct (72B) | 74.57% | 38.36% | 55.37% | 60.94% | 0.864 | 0.851 | 1.860 | 1.348 | 4.923 |
| InternVL3 (8B) | 75.09% | 46.98% | 50.62% | 60.94% | 0.937 | 0.878 | 1.864 | 1.339 | 5.018 |
| InternVL3 (38B) | 70.62% | 47.84% | 51.24% | 59.32% | 0.928 | 0.900 | 1.900 | 1.367 | 5.095 |
| BiMediX2 (8B) | 47.77% | 28.02% | 29.13% | 37.29% | 0.367 | 0.376 | 0.348 | 0.683 | 1.774 |
| MedGemma (27B) | 62.71% | 44.40% | 49.59% | 54.55% | 0.742 | 0.466 | 1.652 | 1.249 | 4.109 |
| Lingshu (32B) | 48.80% | 50.86% | 53.31% | 50.85% | 0.629 | 0.733 | 1.964 | 1.059 | 4.385 |
| Mistral-Medium-3 | 65.46% | 46.12% | 52.89% | 57.32% | 0.937 | 0.805 | 1.652 | 1.389 | 4.783 |
| Claude-4-Sonnet | 67.53% | 39.66% | 53.93% | 57.47% | 0.837 | 0.674 | 1.810 | 1.385 | 4.706 |
| Gemini-2.5-Pro | 70.10% | 52.16% | 46.90% | 58.24% | 0.810 | 0.769 | 1.579 | 1.548 | 4.706 |
| GPT-4o | 73.54% | 48.71% | 52.89% | 61.40% | 0.923 | 0.936 | 1.809 | 1.389 | 5.057 |
| Grok-4 | 76.98% | 46.55% | 63.22% | 66.41% | 1.036 | 0.937 | 1.751 | 1.484 | 5.208 |
| GPT-5 | 78.52% | 57.33% | 56.61% | 66.56% | 1.176 | 1.090 | 1.756 | 1.566 | 5.588 |

Table 6: Performance of different models on perception and no-reference reasoning tasks (Dev Set).

**Few-Shot Evaluation for Context Calibration.** To further calibrate the models' internal quality standards on the No-Reference Reasoning tasks, we additionally perform few-shot evaluations. For each test sample, we provide three carefully selected demonstration examples spanning different imaging modalities and the three quality levels (Good / Usable / Reject). These modality-aware reference examples serve two purposes: *(1)* providing medical context to help models understand how visual degradations influence diagnostic utility across different imaging scenarios; and *(2)* establishing concrete anchors that allow models to interpret subjective quality categories more reliably, rather than relying on abstract definitions alone. Table 7 presents the few-shot evaluation results on the No-Reference Reasoning tasks. Compared with the zero-shot results in Table 1, modality-aware reference examples yield consistent improvements across most evaluation dimensions. These results show that few-shot demonstrations enhance models' interpretation of the quality labels and improve calibration in medical image quality assessment.

**Paired Comparison Task Performance Breakdown.** Table 10 provides the complete numerical breakdown of model performance across different comparison difficulty levels and evaluation dimensions corresponding to Figure 6. This comprehensive analysis reveals significant performance variations between coarse-grained and fine-grained comparison tasks across all models.

---

[3]https://github.com/open-compass/VLMEvalKit

| Model | Comp.↑ | Prec.↑ | Cons.↑ | Qual.↑ | Overall↑ |
|---|---|---|---|---|---|
| Qwen2.5-VL-Instruct (7B) | 0.769 | 0.723 | 1.878 | 1.168 | 4.538 |
| Qwen2.5-VL-Instruct (32B) | 1.082 | 0.959 | **1.982** | 1.335 | 5.358 |
| InternVL3 (8B) | 1.005 | 0.950 | 1.905 | 1.421 | 5.281 |
| InternVL3 (38B) | 0.996 | 0.973 | 1.932 | 1.448 | 5.349 |
| Qwen2.5-VL-Instruct (72B) | 0.932 | 0.923 | 1.896 | 1.412 | 5.163 |
| BiMediX2 (8B) | 0.405 | 0.428 | 0.315 | 0.715 | 1.863 |
| Lingshu (32B) | 0.678 | 0.751 | 1.955 | 1.105 | 4.489 |
| MedGemma (27B) | 0.796 | 0.523 | 1.687 | 1.314 | 4.320 |
| Mistral-Medium-3 | 0.987 | 0.878 | 1.742 | 1.445 | 5.052 |
| Claude-4-Sonnet | 0.914 | 0.787 | 1.869 | 1.476 | 5.046 |
| Gemini-2.5-Pro | 0.905 | 0.860 | 1.710 | 1.605 | 5.080 |
| Grok-4 | 1.095 | 1.005 | 1.823 | 1.550 | 5.473 |
| GPT-4o | 1.077 | 1.036 | 1.860 | 1.520 | 5.493 |
| GPT-5 | **1.220** | **1.145** | 1.850 | **1.630** | **5.845** |

Table 7: Few-shot evaluation results on No-reference Reasoning tasks.

| Model | CT | Histo. | MRI | Endos. | Retinal |
|---|---|---|---|---|---|
| GPT-5 | 71.47% | **65.43%** | **75.90%** | 60.89% | **70.09%** |
| GPT-4o | **72.85%** | 58.33% | 64.75% | 60.44% | 66.67% |
| Grok-4 | 70.14% | 59.37% | 65.93% | 64.49% | 61.40% |
| Gemini-2.5-Pro | 67.04% | 53.40% | 60.79% | 60.89% | 59.83% |
| Mistral-Medium-3 | 65.93% | 38.58% | 61.51% | **65.33%** | 61.54% |
| Claude-4-Sonnet | 64.27% | 54.63% | 55.04% | **65.33%** | 65.81% |
| Qwen2.5-VL-72B | 65.65% | 47.53% | 74.82% | 66.22% | 64.96% |
| InternVL3-38B | 68.14% | 48.46% | 62.95% | 60.44% | **70.09%** |
| InternVL3-8B | 60.66% | 51.54% | 61.87% | 67.56% | 63.25% |
| Qwen2.5-VL-32B | 59.00% | 46.30% | 66.55% | 67.56% | 63.25% |
| Qwen2.5-VL-7B | 56.79% | 35.49% | 65.47% | 60.44% | 64.96% |
| MedGemma-27B | 66.57% | 46.60% | 57.55% | 56.00% | 59.83% |
| Lingshu-32B | 57.89% | 35.19% | 61.15% | 50.67% | 48.72% |
| BiMediX2-8B | 41.99% | 23.38% | 44.36% | 38.74% | 47.62% |

Table 8: Detailed perception accuracy results across five imaging modalities on the test set.

| Model | Comp. | Prec. | Cons. | Qual. | Overall |
|---|---|---|---|---|---|
| GPT-5 | **1.376** | **1.504** | 1.895 | **1.609** | **6.384** |
| GPT-4o | 1.113 | 1.489 | **1.947** | 1.669 | 6.218 |
| Grok-4 | 1.203 | 1.203 | 1.865 | 1.421 | 5.692 |
| Gemini-2.5-Pro | 1.008 | 1.180 | 1.895 | 1.489 | 5.572 |
| Mistral-Medium-3 | 0.932 | 1.263 | 1.789 | 1.414 | 5.398 |
| Claude-4-Sonnet | 0.827 | 0.992 | 1.917 | 1.338 | 5.074 |
| Qwen2.5-VL-72B-Instruct | 0.947 | 1.158 | 1.481 | 1.376 | 4.962 |
| InternVL3-38B | 1.090 | 1.090 | 1.684 | 1.489 | 5.353 |
| InternVL3-8B | 1.023 | 1.278 | 1.910 | 1.549 | 5.760 |
| Qwen2.5-VL-32B-Instruct | 0.865 | 0.872 | 1.887 | 1.083 | 4.707 |
| Qwen2.5-VL-7B-Instruct | 0.684 | 0.925 | 1.316 | 1.150 | 4.075 |
| MedGemma-27B | 0.662 | 0.571 | 1.105 | 0.955 | 3.293 |
| Lingshu-32B | 0.692 | 0.940 | 1.519 | 1.203 | 4.354 |
| BiMediX2-8B | 0.526 | 0.579 | 0.594 | 0.511 | 2.210 |

Table 9: Performance comparison on MedQ-Reasoning paired comparison tasks (Dev Set).

| Model | Group | Comp. | Prec. | Cons. | Qual. Acc. | Total |
|---|---|---|---|---|---|---|
| GPT-5 | Overall | 1.293 | 1.556 | 1.925 | 1.564 | 6.338 |
| | Fine-grained | 1.301 | 1.495 | 1.903 | 1.476 | 6.175 |
| | Coarse-grained | 1.267 | 1.767 | 2.000 | 1.867 | 6.900 |
| GPT-4o | Overall | 1.105 | 1.414 | 1.632 | 1.564 | 5.714 |
| | Fine-grained | 1.155 | 1.388 | 1.583 | 1.534 | 5.660 |
| | Coarse-grained | 0.933 | 1.500 | 1.800 | 1.667 | 5.900 |
| Grok-4 | Overall | 1.150 | 1.233 | 1.820 | 1.459 | 5.662 |
| | Fine-grained | 1.214 | 1.350 | 1.825 | 1.495 | 5.883 |
| | Coarse-grained | 0.933 | 0.833 | 1.800 | 1.333 | 4.900 |
| Gemini-2.5-Pro | Overall | 1.053 | 1.233 | 1.774 | 1.534 | 5.594 |
| | Fine-grained | 1.039 | 1.262 | 1.709 | 1.476 | 5.485 |
| | Coarse-grained | 1.100 | 1.133 | 2.000 | 1.733 | 5.967 |
| InternVL3-8B | Overall | 0.985 | 1.278 | 1.797 | 1.474 | 5.534 |
| | Fine-grained | 0.971 | 1.155 | 1.748 | 1.359 | 5.233 |
| | Coarse-grained | 1.033 | 1.700 | 1.967 | 1.867 | 6.567 |
| Claude-4-Sonnet | Overall | 0.857 | 1.083 | 1.910 | 1.481 | 5.331 |
| | Fine-grained | 0.835 | 1.107 | 1.883 | 1.495 | 5.320 |
| | Coarse-grained | 0.933 | 1.000 | 2.000 | 1.433 | 5.367 |
| Mistral-Medium-3 | Overall | 0.872 | 1.203 | 1.827 | 1.338 | 5.241 |
| | Fine-grained | 0.893 | 1.252 | 1.786 | 1.282 | 5.214 |
| | Coarse-grained | 0.800 | 1.033 | 1.967 | 1.533 | 5.333 |
| InternVL3-38B | Overall | 1.075 | 1.083 | 1.571 | 1.414 | 5.143 |
| | Fine-grained | 1.117 | 1.184 | 1.466 | 1.359 | 5.126 |
| | Coarse-grained | 0.933 | 0.733 | 1.933 | 1.600 | 5.200 |
| Lingshu-32B | Overall | 0.729 | 1.015 | 1.586 | 1.323 | 4.654 |
| | Fine-grained | 0.699 | 0.990 | 1.505 | 1.243 | 4.437 |
| | Coarse-grained | 0.833 | 1.100 | 1.867 | 1.600 | 5.400 |
| Qwen2.5-VL-32B | Overall | 0.692 | 0.752 | 1.895 | 0.962 | 4.301 |
| | Fine-grained | 0.786 | 0.922 | 1.864 | 1.068 | 4.641 |
| | Coarse-grained | 0.367 | 0.167 | 2.000 | 0.600 | 3.133 |
| Qwen2.5-VL-7B | Overall | 0.714 | 0.902 | 1.316 | 1.143 | 4.075 |
| | Fine-grained | 0.757 | 1.000 | 1.320 | 1.175 | 4.252 |
| | Coarse-grained | 0.567 | 0.567 | 1.300 | 1.033 | 3.467 |
| Qwen2.5-VL-72B | Overall | 0.737 | 0.977 | 1.233 | 1.113 | 4.060 |
| | Fine-grained | 0.699 | 0.903 | 1.029 | 0.971 | 3.602 |
| | Coarse-grained | 0.867 | 1.233 | 1.933 | 1.600 | 5.633 |
| MedGemma-27B | Overall | 0.684 | 0.692 | 1.128 | 1.000 | 3.504 |
| | Fine-grained | 0.650 | 0.641 | 0.942 | 0.854 | 3.087 |
| | Coarse-grained | 0.800 | 0.867 | 1.767 | 1.500 | 4.933 |
| BiMediX2-8B | Overall | 0.474 | 0.549 | 0.639 | 0.511 | 2.173 |
| | Fine-grained | 0.359 | 0.379 | 0.641 | 0.311 | 1.689 |
| | Coarse-grained | 0.867 | 1.133 | 0.633 | 1.200 | 3.833 |

Table 10: Detailed numerical results for paired comparison reasoning tasks across models, corresponding to Figure 6.

## A.5 QUALITATIVE ANALYSIS AND CASE STUDIES

### A.5.1 WHY DO MEDICAL-SPECIALIZED MODELS UNDERPERFORM GENERAL-PURPOSE MODELS?

The counterintuitive finding that medical-specialized models consistently underperform general-purpose models across all evaluation dimensions warrants comprehensive analysis. Figure 11 provides illustrative examples demonstrating fundamental limitations in medical-specialized models' low-level visual perception capabilities.

**Insufficient Low-Level Visual Attribute Training.** Medical-specialized models appear to prioritize high-level diagnostic reasoning over fundamental visual perception skills. In the CT scan example (Figure 11), MedGemma-27B correctly identifies anatomical structures and acknowledges the presence of streak artifacts, but fails to appropriately assess their clinical significance. The model describes the image as "usable but not optimal" despite prominent metal artifacts that would necessitate repeat scanning in clinical practice. This suggests that medical fine-tuning datasets may inadequately represent the full spectrum of image quality degradations encountered in clinical workflows.

**Diagnostic Bias Over Quality Assessment.** BiMediX2-8B demonstrates a critical failure mode by describing the same severely degraded CT scan as having "good quality and suitable for diagnosis." This systematic misalignment indicates that medical-specialized training may inadvertently optimize models for diagnostic confidence rather than quality assessment accuracy. The model's focus on anatomical identification overshadows its ability to detect quality-compromising artifacts, suggesting that current medical training paradigms may not adequately distinguish between diagnostic content recognition and image quality evaluation.

### A.5.2 ERROR PATTERN ANALYSIS OF TOP-PERFORMING MODELS

To identify which inputs contribute most to errors in state-of-the-art MLLMs, we analyzed GPT-5's failure patterns across different modalities and artifact types on the perception tasks. Table 11 shows that the highest error rates appear in Endoscopy (39.11%) and Histopathology (34.57%) modalities. These modalities contain complex textures, heterogeneous visual structures, and tissue-level variations that increase the difficulty of fine-grained quality assessment compared to volumetric imaging modalities such as CT and MRI.

At the artifact level, the most challenging categories for GPT-5 are Air Bubbles (95.23% error rate), Specularity (90.48%), and Blood-related artifacts (84.62%). These failures arise because such artifacts exhibit irregular spatial distributions, high-frequency visual patterns, and often co-occur with other degradations. Notably, these challenging artifacts predominantly appear in endoscopy and histopathology images, where strong localized degradations and complex tissue backgrounds make accurate identification particularly difficult even for top-performing models. This analysis reveals that current MLLMs still struggle with modality-specific artifacts that require fine-grained visual discrimination in texturally complex medical imaging contexts.

| Modality | Error Rate |
|---|---|
| Endoscopy | 39.11% |
| Histopathology | 34.57% |
| Fundus | 29.91% |
| CT | 25.76% |
| MRI | 24.10% |

Table 11: GPT-5 error rates across imaging modalities on perception tasks, ranked by difficulty.

### A.5.3 EXAMPLE OF REASONING TASKS

**Question**: Is there a ring artifact visible in the CT image of the abdomen?
A. Yes, B. No.
**Correct Answer**: A

(a)

✔ **InternVL3-8B**: A
✔ **Qwen2.5-VL-32B-Instruct**: A.
✘ **Lingshu-32B**: B.
✘ **mistral-medium-3_trans**: B
✔ **BiMediX2-8B**: A
✔ **GPT4o_trans**: A
✔ **Claude4_Sonnet_trans**: A
✔ **InternVL3-38B**: A
✔ **gpt-5**: A
✘ **Qwen2.5-VL-72B-Instruct**: B
✘ **Qwen2.5-VL-7B-Instruct**: B.
✔ **MedGemma-27B**: A
✔ **grok-4**: A
✔ **GeminiPro2-5_trans**: A

**Question**: What is the primary quality issue of this image?
A. Blurry and out of focus, B. Poor contrast and tonal separation,
C. Excessive noise and artifacts, D. Insufficient illumination
**Correct Answer**: D

(b)

✘ **InternVL3-8B**: B
✔ **Qwen2.5-VL-32B-Instruct**: D
✘ **Lingshu-32B**: B.
✘ **mistral-medium-3_trans**: B. Poor contrast and tonal separation
✘ **BiMediX2-8B**: B
✔ **GPT4o_trans**: D
✔ **Claude4_Sonnet_trans**: D
✘ **InternVL3-38B**: A
✔ **gpt-5**: D
✔ **Qwen2.5-VL-72B-Instruct**: D
✔ **Qwen2.5-VL-7B-Instruct**: D
✔ **MedVLM-R1**: D
✔ **grok-4**: D
✔ **GeminiPro2-5_trans**: D

Figure 10: Representative QA examples demonstrating typical question-answer patterns in MedQ-Bench across different medical imaging modalities and quality assessment scenarios.

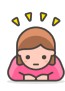 **User:** As a medical image quality assessment expert, provide a concise description focusing on low-level appearance of the image in details. Conclude with "Overall, the quality of this image is [good/usable/reject]".

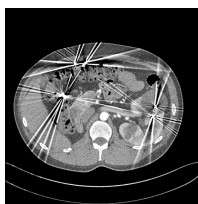 **Golden Reasoning:** This is a CT scan of the abdominal region. The image exhibits moderate clarity and contrast, but the presence of significant noise reduces overall resolution. Metal artifacts are prominent, creating streaks that obscure anatomical details and hinder accurate interpretation of the abdominal structures. Overall, the quality of this image is reject.

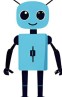 **InternVL3-8B:** This abdominal CT scan shows severe streak artifacts radiating from the center, likely due to beam hardening or metal. While overall contrast is adequate and major structures like vertebrae and soft tissues are visible, the artifacts obscure finer details and reduce clarity. Some noise is present but less problematic than the artifacts. Overall, the image quality is poor and judged as reject.

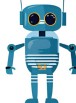 **MedGemma-27B:** This abdominal CT scan shows bones, soft tissues, and air with adequate brightness and contrast. However, streak artifacts from dense structures, such as bone or metal, obscure fine details. Moderate noise adds a grainy texture, further reducing clarity. While the image remains interpretable for basic anatomical identification, the artifacts and noise limit diagnostic precision. Overall, the quality is usable but not optimal.

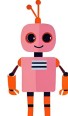 **Lingshu-32B:** This CT scan of the abdomen or pelvis shows the vertebral column, soft tissues, and possible lesions. However, it is heavily degraded by bright streak artifacts that obscure much of the anatomy. These distortions suggest poor acquisition parameters or technical issues. As a result, the image is of low quality and not reliable for diagnostic purposes.

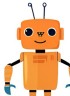 **BiMediX2-8B:** This CT scan of the liver shows clear structures with good contrast and minimal noise. The organ and surrounding tissues are well-defined, supporting accurate anatomical assessment. No significant artifacts are visible, and the overall resolution is sufficient for clinical use. Overall, the image quality is good and suitable for diagnosis.

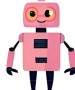 **GPT-4o:** The abdominal CT image contains multiple bright streak artifacts, likely from metal or dense materials. These streaks obscure anatomical details and reduce clarity. Uneven brightness and contrast further complicate interpretation. Due to the severity of these distortions, the overall image quality is reject.

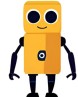 **Claude4 Sonnet:** This CT scan is heavily degraded by extensive streak artifacts radiating across the field of view, creating a star-like pattern. The artifacts, likely from metallic implants, obscure soft tissue structures and organ boundaries. Contrast is poor, noise is elevated, and diagnostic interpretation is nearly impossible in affected regions. While a few landmarks remain visible, the image quality overall is reject.

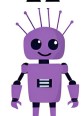 **InternVL3-38B:** This abdominal CT scan shows vertebrae, soft tissues, and organs with adequate contrast and sharpness. However, background noise is high, which reduces the clarity of smaller structures and subtle abnormalities. No major artifacts obstruct critical regions, and the field of view is appropriate. Overall, the image is usable but limited by noise.

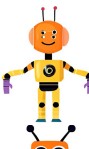 **GPT-5:** This axial CT image shows moderate noise and acceptable sharpness, with organ edges reasonably defined. However, severe metal-induced streak artifacts radiate across the field, producing alternating bright and dark bands that obscure anatomy. Global shading and banding further reduce uniformity. While centered and properly cropped, the distortions make diagnostic interpretation unreliable. Overall, the quality is reject.

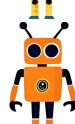 **Qwen2.5-VL-72B-Instruct:** The CT scan is degraded by strong streak artifacts, likely from metallic implants or dense objects. These streaks obscure anatomical details and compromise evaluation of affected regions. Contrast and brightness remain acceptable in unaffected areas, preserving partial visibility of tissues. Despite this, the diagnostic utility is poor. Overall, the quality is reject.

Figure 11: Representative no-reference reasoning image examples demonstrating typical question-answer patterns in MedQ-Bench across different models.

**User:** As a medical image quality assessment expert, provide a concise description comparing two images focusing on low-level appearance. Conclude with which image has higher quality.

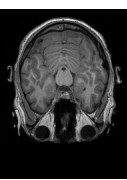 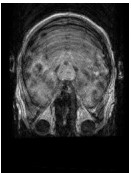

**Golden Reasoning:** The first image demonstrates excellent clarity and sharpness with minimal motion artifacts, allowing for clear visualization of anatomical structures. In contrast, the second image exhibits severe motion artifacts, including blurring and ghosting, which significantly obscure anatomical details and reduce diagnostic quality. The brain tissue boundaries and ventricular structures are clearly visible in the first image, while they are severely degraded in the second. Overall, the first image has higher quality.

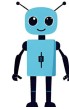 **InternVL3-8B:** The comparison of the two images shows clear quality differences. Image 1 has better contrast and clarity, with gray and white matter well-differentiated and minimal noise, making structures like gyri and sulci more distinct. Its resolution is higher, and artifacts are minimal, ensuring reliable diagnostic value. Image 2, by contrast, suffers from lower contrast and clarity, higher noise, and poorer resolution, making brain structures harder to interpret. It also contains more artifacts, which further obscure details. Conclusion: Image 1 offers superior diagnostic quality, providing clearer visualization of anatomical structures.

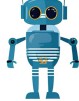 **MedGemma-27B:** Image 1 shows clearer brain structures with better gray–white matter contrast and less noise. Image 2 appears blurrier, with weaker contrast and more graininess, making details harder to identify. Overall, Image 1 demonstrates superior clarity and quality.

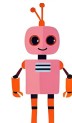 **Lingshu-32B:** The first image is a coronal T1-weighted MRI with clear anatomical structures, good signal-to-noise ratio, and strong contrast between gray and white matter. These qualities make it suitable for accurate interpretation. The second image, however, is heavily corrupted by noise, which obscures anatomical details and reduces clarity. The poor visibility significantly limits its diagnostic value. Overall, the first image is of much higher quality.

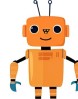 **BiMediX2-8B:** The left image demonstrates a higher signal-to-noise ratio, producing a clearer and more detailed view of anatomical structures. In contrast, the right image shows lower SNR, resulting in reduced clarity and less visible details. This difference strongly impacts the ability to detect and diagnose subtle abnormalities. Overall, the left image provides superior quality and greater diagnostic reliability.

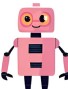 **GPT-4o:** The first image shows higher contrast, clearer anatomical structures, and less noise, with well-defined edges. The second image has lower contrast and more noise, causing structures to appear blurred and less distinct. Overall, the first image demonstrates superior quality due to its clarity and better contrast.

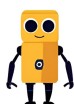 **Claude4 Sonnet:** The left MRI image has higher contrast, sharper anatomical boundaries, and a better signal-to-noise ratio, with clear visualization of brain parenchyma and ventricles. Its signal is more uniform and details are well-preserved. The right image, however, appears grainier with lower contrast, more noise, and blurred structures, showing reduced diagnostic value. Overall, the left image is of higher quality and more suitable for clinical interpretation.

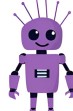 **InternVL3-38B:** The first MRI image shows clear brain structures with distinct tissue contrasts and high resolution, making fine details easy to visualize. In comparison, the second image appears lower in quality, with weaker tissue contrast and less distinct details. Its resolution is reduced, making fine anatomical features harder to discern. Overall, the first image demonstrates higher quality due to sharper details and better contrast.

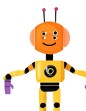 **GPT-5:** The left image has sharp cortical and orbital boundaries, low noise, and strong gray–white matter contrast. Its background is smooth and uniform, with minimal artifacts. By contrast, the right image is blurred, noisy, and shows reduced contrast with visible mottling and ghosting. These issues obscure fine details and reduce diagnostic value. Overall, the left image is of clearly higher quality.

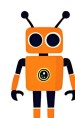 **Qwen2.5-VL-72B-Instruct:** Image 1 has lower contrast, less differentiation between tissues, and smoother appearance with reduced sharpness. Image 2 shows higher contrast and clearer anatomical details, though it also contains more visible noise. Despite the noise, the sharper features and stronger tissue contrast make Image 2 more suitable for diagnostic use. Overall, Image 2 is of higher quality.

Figure 12: Representative paired image examples demonstrating typical question-answer patterns in MedQ-Bench across different models.

### A.5.4 Human Expert Evaluation Protocol

**Expert Recruitment and Qualification Criteria.** Human experts in our evaluation consisted of medical imaging technicians with a minimum of 3 years of clinical experience in medical imaging quality assessment and medical imaging PhDs with specialized training in image quality evaluation. Medical imaging technicians were recruited from certified clinical facilities and possessed active professional certifications in their respective imaging modalities. PhDs were selected from accredited medical imaging research programs and had completed at least 2 years of coursework, including medical image processing and quality assessment methodologies. All experts demonstrated proficiency in identifying common imaging artifacts and quality issues across multiple medical imaging modalities through a standardized pre-evaluation assessment.

**Human-Human Annotation Alignment Analysis.** To quantify the reliability achieved by our multi-round expert validation protocol, we computed Cohen's $\kappa$ between each annotator and the consensus label obtained via majority voting. For the reasoning tasks, Table 12 presents the inter-annotator agreement statistics. For the perception tasks, agreement between individual annotators and the majority-vote labels was $\kappa = 0.873$. These $\kappa$ values indicate substantial to almost perfect agreement under our structured expert protocol, confirming that the annotation process minimizes bias and ensures reliability.

| Metric | Completeness | Preciseness | Consistency | Quality Accuracy |
|---|---|---|---|---|
| Cohen's $\kappa$ | 0.843 | 0.817 | 0.856 | 0.884 |

Table 12: Inter-annotator agreement (Cohen's $\kappa$) for reasoning task.

**Human-AI Alignment Analysis.** The confusion matrices shown in Figure 13 demonstrate strong alignment between human expert scores and GPT-4o automated evaluation across all three evaluation dimensions, with over 80% accuracy in each dimension. Quadratic weighted $\kappa_w$ accounts for the ordinal nature of the evaluation labels, penalizing larger discrepancies more heavily than adjacent category differences. The consistently high $\kappa_w$ values (0.774–0.985) detailed in Table 13 indicate substantial agreement beyond chance between human expert scores and GPT-4o automated evaluation, reflecting that the automated system is not only accurate but also aligned with the fine-grained ordinal structure of human expert judgments.

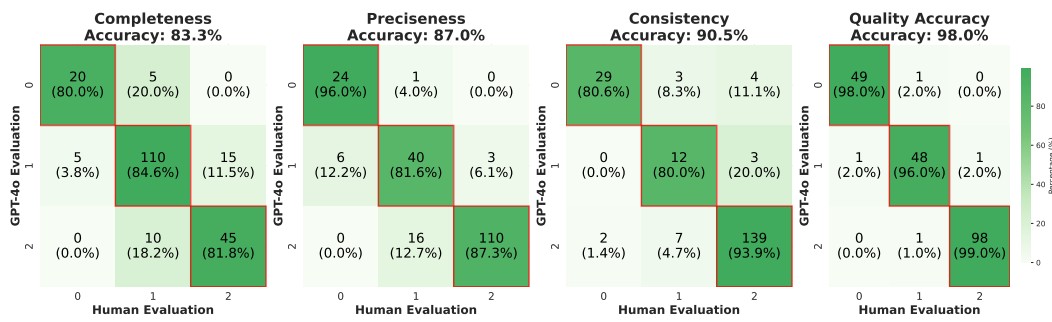

Figure 13: Confusion matrices showing alignment between human expert scores and GPT-4o automated evaluation across four evaluation dimensions.

| Metric | Completeness | Preciseness | Consistency | Quality Accuracy |
|---|---|---|---|---|
| $\kappa_w$ | 0.774 | 0.876 | 0.840 | 0.985 |

Table 13: Quadratic weighted Cohen's $\kappa_w$ values for human–AI alignment across evaluation dimensions.

To assess whether human–AI alignment remains stable across quality levels, we partitioned the 200 sampled outputs into three tiers by human-score quantiles and examined the lowest third, which contains more low-quality or failed predictions. Table 14 presents the alignment metrics in this challenging subset. While the weighted $\kappa_w$ values show some decrease compared to overall alignment, the automated evaluation does not substantially degrade even under these challenging cases, with $\kappa_w$ values remaining above 0.7 for most dimensions and achieving 0.952 for Quality Accuracy. This confirms that our automated evaluation framework maintains robust alignment with human expert annotations, strengthening confidence in its use as a reliable surrogate for large-scale human evaluation.

| Metric | Completeness | Preciseness | Consistency | Quality Accuracy |
|---|---|---|---|---|
| $\kappa_w$ | 0.702 | 0.791 | 0.755 | 0.952 |

Table 14: Human–AI alignment in the lowest-quality tier (bottom third by human score). Despite challenging cases, alignment remains robust across all evaluation dimensions.

**Multi-Judge Validation for Self-Preference Analysis.** To investigate potential self-preference bias when using GPT-4o as the evaluation judge, we conducted a multi-judge validation study using three independent LLM judges: GPT-4o, Claude-4-Sonnet, and Gemini-2.5-Pro. Each judge independently scored all model outputs on the No-Reference Reasoning tasks. Table 15 presents the overall scores assigned by each judge and inter-judge consistency metrics. The results show minimal variation between judges, with score differences typically within 0.05 points. Notably, GPT-4o does not assign systematically higher scores to OpenAI models (GPT-4o, GPT-5) compared to scores from Claude or Gemini judges. All models achieve Intraclass Correlation Coefficient (ICC) values above 0.95, indicating excellent agreement among judges. The mean absolute deviation (MAD) between GPT-4o scores and other judges remains below 0.05 for most models, confirming that GPT-4o's evaluations are consistent with independent judges and do not exhibit self-preference bias.

| Model | GPT-4o | Claude | Gemini | ICC | ICC 95% CI | MAD (Claude) | MAD (Gemini) |
|---|---|---|---|---|---|---|---|
| Qwen2.5-VL-Instruct (7B) | 4.367 | 4.376 | 4.371 | 0.984 | [0.980, 0.987] | 0.009 | 0.004 |
| Qwen2.5-VL-Instruct (32B) | 5.272 | 5.294 | 5.308 | 0.992 | [0.990, 0.994] | 0.022 | 0.036 |
| InternVL3 (8B) | 4.983 | 4.995 | 4.968 | 0.986 | [0.982, 0.989] | 0.012 | 0.015 |
| InternVL3 (38B) | 4.965 | 4.977 | 4.977 | 0.990 | [0.988, 0.992] | 0.012 | 0.012 |
| Qwen2.5-VL-Instruct (72B) | 4.982 | 4.932 | 4.959 | 0.988 | [0.985, 0.990] | 0.050 | 0.023 |
| BiMediX2 (8B) | 1.721 | 1.692 | 1.719 | 0.954 | [0.942, 0.963] | 0.029 | 0.002 |
| Lingshu (32B) | 4.312 | 4.344 | 4.317 | 0.987 | [0.984, 0.990] | 0.032 | 0.005 |
| MedGemma (27B) | 4.054 | 4.036 | 4.036 | 0.978 | [0.973, 0.983] | 0.018 | 0.018 |
| Mistral-Medium-3 | 4.557 | 4.529 | 4.520 | 0.988 | [0.985, 0.990] | 0.028 | 0.037 |
| Claude-4-Sonnet | 4.529 | 4.529 | 4.570 | 0.981 | [0.976, 0.985] | 0.000 | 0.041 |
| Gemini-2.5-Pro | 5.018 | 5.014 | 5.005 | 0.985 | [0.981, 0.988] | 0.004 | 0.013 |
| GPT-4o | 5.321 | 5.335 | 5.321 | 0.989 | [0.986, 0.991] | 0.014 | 0.000 |
| GPT-5 | 5.679 | 5.679 | 5.652 | 0.987 | [0.983, 0.989] | 0.000 | 0.027 |

Table 15: Multi-judge validation results showing overall reasoning scores from three independent LLM judges and inter-judge consistency metrics. MAD (Claude) and MAD (Gemini) represent MAD between GPT-4o scores and Claude/Gemini scores respectively. Score differences across judges are minimal, and high ICC values ($>0.95$) confirm excellent agreement, indicating no systematic self-preference bias.

## A.6 DISCUSSION

**Task-Agnostic vs. Task-Specific Quality Assessment.** We acknowledge that medical image quality can depend on the specific diagnostic task in clinical practice, where the same image may be satisfactory for one diagnostic purpose but inadequate for another. In this work, however, we intentionally focus on task-agnostic and foundational quality perception, which allows us to evaluate models' low-level visual understanding without mixing it with task-specific diagnostic reasoning. To keep the evaluation clinically meaningful within this scope and to minimize subjectivity and ensure fair comparison across models, we adopt two complementary strategies. First, we use strict system

prompts that define a shared scoring rubric for all models. Before answering each question, every model receives rigorous, structured instructions that specify the Good, Usable, and Reject quality levels and the required low-level visual criteria, which correspond to common low-level attributes that matter across diagnostic tasks. These definitions are aligned with expert annotations through our multi-round validation protocol (Appendix A.1), ensuring that all evaluated LLMs operate under the same medically grounded scoring system. By focusing on fundamental quality attributes that affect general medical image usability, we can systematically examine whether MLLMs possess the core perceptual capabilities needed to recognize and reason about quality degradations across diverse medical imaging modalities. MedQ-Bench provides the essential foundation for evaluating low-level quality perception capabilities rather than a complete end-to-end validation of clinical deployment safety. Extending the benchmark toward context-dependent, task-aware IQA represents an important direction for future studies building upon this foundation.

**Systematic Failure Mode Analysis.** Our evaluation reveals four primary failure patterns in current MLLMs with concrete evidence. First, *insufficient low-level attribute understanding*, particularly for medical-specific quality attributes: current models show limited accuracy on perception tasks (Table 1), with modality-specific questions performing substantially worse than general questions (Figure 5b), suggesting that training data lacks comprehensive low-level visual attribute descriptions tailored to medical imaging modalities. Second, *quantitative weaknesses in fine-grained distinctions*: models struggle significantly with mild degradation cases compared to severe degradations (Figure 5a), achieving only 56% accuracy on mild degradations and showing substantial performance gaps between fine-grained and coarse-grained comparisons (Table 10), reflecting challenges in subtle visual sensitivity. Third, *limited reasoning capabilities*: the reasoning deficiencies stem from both inadequate perceptual foundation and insufficient reasoning mechanisms, as evidenced by low completeness and preciseness scores in reasoning tasks (Table 1), where models struggle to provide comprehensive quality descriptions and maintain logical consistency between observations and conclusions. Fourth, *diagnostic bias in medical-specialized models*: systems such as BiMediX2-8B overrate severely degraded images (describing them as "good quality and suitable for diagnosis", Figure 11) because their training data emphasizes disease recognition over quality supervision, conflating diagnostic recognizability with image quality.

**Technical Implications for MLLM Development.** Based on these findings, we identify four concrete directions for improving medical MLLMs. First, *modality-aware quality training*: future systems should incorporate comprehensive low-level attribute descriptions for medical-specific degradations during training, addressing the knowledge gaps revealed by poor performance on modality-specific quality attributes. Second, *fine-grained perceptual enhancement*: degradation-focused pre-training or synthetic artifact augmentation targeting mild degradations can improve sensitivity to subtle defects, addressing the substantial performance gaps between fine-grained and coarse-grained quality distinctions. Third, *structured quality reasoning alignment*: reinforcement learning methods such as GRPO (Group Relative Policy Optimization) can enhance reasoning capabilities by providing explicit reward signals based on quality assessment accuracy, addressing the insufficient reasoning mechanisms revealed in our evaluation. Fourth, *quality-diagnosis decoupling*: medical models should explicitly separate low-level artifact modeling from diagnostic reasoning through degradation-aware training objectives, preventing the conflation between recognizability and image quality observed in current medical-specialized systems.

