# OpenReview forum: "MedQ-Bench: Evaluating and Exploring Medical Low-level Visual Abilities in MLLMs"
_ICLR.cc/2026/Conference — ICLR 2026 Conference Desk Rejected Submission_

### Official Review · Reviewer_wTsk · 2025-10-21

**Soundness:** 3
**Presentation:** 3
**Contribution:** 3
**Rating:** 6
**Confidence:** 4

**Summary:**

This paper proposes a benchmarking framework for probing the capabilities of multi-modal LLMs on medical image quality assessment. Existing evaluation metrics are scalar, score-based and cannot capture the nuances of human expert evaluations of such images. The paper defines two main tasks: 1) simple perception-based questions on image artifacts, degradations and other low-level visual characteristics, and 2) a reasoning task more akin to expert evaluation. Authors introduce a judging protocol that assesses the model outputs along different dimensions, where alignment with human judgment is demonstrated. State-of-the-art models demonstrate performance somewhere between a non-expert and expert human, suggesting the need for further improvements in the medical domain.

**Strengths:**

- The paper has strong motivation and it is presented well. I especially like Fig. 2 demonstrating how traditional image similarity metrics fail to capture medical image quality for diagnostic purposes.

- The benchmarking framework is overall reasonable, it incorporates a good mix of fundamental image quality assessment problems and more advanced reasoning tasks.

- Questions targeting imaging artifacts are tailored towards the specific medical imaging modalities.

- The benchmark and evaluation protocol incorporated human experts in multiple stages, supporting the reliability of the benchmark.

**Weaknesses:**

- One key aspect I believe this benchmark is missing is the context-dependence of medical image quality. Depending on the diagnostic task, the same image can be of satisfactory for one task and unusable for another. To my understanding, this benchmark cannot consider this aspect as the questions are directly asking about an absolute score of image quality. Adding medical context to the QA problems would address this issue and make the benchmark more complete.

- It appears to me that some evaluation dimensions can be subjective (e.g. no/mild/severe degradation) and might confuse the LLM. Without some kind of reference it can be challenging for the model to determine if a degraded image is mildly or severely degraded outside of medical context. One way to address this would be few-shot evaluations, where the model is shown some demonstrations before asking each of the questions and thus can "calibrate" its responses.

- A more in-depth analysis of concrete failure modes would be helpful with some discussion on how to address the shortcomings of current models, highlighting potential future directions in model design or training improvements.

**Questions:**

- How does the current benchmark consider the medical context when assessing image quality?

- How is it ensured that the evaluated LLMs understand the scoring system introduced by the paper for fair/comparable evaluation?

- What are the technical implications of the paper on the design and development of medical MLLMs?

---

> ### Author Response · Authors · 2025-11-23
> **Response to Reviewer wTsk (1/3)**
>
> Thank you for your valuable feedback to help us improve our paper. We have revised our paper based on your feedback. We detail our response below, and please kindly let us know if our response addresses your concerns.
>
> ---
> > - W1: One key aspect I believe this benchmark is missing is the context-dependence of medical image quality. Depending on the diagnostic task, the same image can be of satisfactory for one task and unusable for another. To my understanding, this benchmark cannot consider this aspect as the questions are directly asking about an absolute score of image quality. Adding medical context to the QA problems would address this issue and make the benchmark more complete.
> > - Q1: How does the current benchmark consider the medical context when assessing image quality?
> > - Q2: How is it ensured that the evaluated LLMs understand the scoring system introduced by the paper for fair/comparable evaluation?
>
> **A1:** We understand that diagnostic context can influence how clinicians judge image acceptability. In this work, however, we intentionally focus on task-agnostic and foundational quality perception, which allows us to evaluate models’ low-level visual understanding without mixing it with task-specific diagnostic reasoning. To keep the evaluation clinically meaningful within this scope and to minimize subjectivity and ensure fair comparison across models, we adopt two complementary strategies. First, we use strict system prompts that define a shared scoring rubric for all models. Before answering each question, every model receives a rigorous, structured instruction that specifies the Good, Usable, and Reject quality levels and the required low-level visual criteria, which correspond to common low-level attributes that matter across diagnostic tasks. These definitions are aligned with expert annotation, ensuring that all evaluated LLMs operate under the same medically grounded scoring system. While the benchmark does not assign task-specific labels, extending MedQ-Bench toward context-dependent, task-aware IQA is an important direction, and we have added the corresponding discussion in Appendix A.6.

---

> ### Author Response · Authors · 2025-11-23
> **Response to Reviewer wTsk (2/3)**
>
> > - W2: It appears to me that some evaluation dimensions can be subjective (e.g. no/mild/severe degradation) and might confuse the LLM. Without some kind of reference it can be challenging for the model to determine if a degraded image is mildly or severely degraded outside of medical context. One way to address this would be few-shot evaluations, where the model is shown some demonstrations before asking each of the questions and thus can "calibrate" its responses.
>
> **A2:** We perform few-shot evaluations to further calibrate the models’ internal quality standards on the no-reference reasoning tasks. For each test sample, we provide three carefully selected examples spanning different imaging modalities and the three quality levels. As shown in Table T1, modality-aware reference examples lead to consistent improvements over the zero-shot setting, indicating that few-shot demonstrations help models better interpret the quality labels. We have revised the paper accordingly and added these details in Appendix A.4.
>
> **Table T1:** Few-shot Evaluation Results on No-reference Reasoning Tasks
>
> | Model                              | Comp.↑ | Prec.↑ | Cons.↑ | Qual.↑ | Overall↑ |
> |-----------------------------------|--------|--------|--------|--------|----------|
> | Qwen2.5-VL-Instruct (7B)           | 0.769  | 0.723  | 1.878  | 1.168  | 4.538    |
> | Qwen2.5-VL-Instruct (32B)          | 1.082  | 0.959  | **1.982** | 1.335  | 5.358    |
> | InternVL3 (8B)                     | 1.005  | 0.950  | 1.905  | 1.421  | 5.281    |
> | InternVL3 (38B)                    | 0.996  | 0.973  | 1.932  | 1.448  | 5.349    |
> | Qwen2.5-VL-Instruct (72B)          | 0.932  | 0.923  | 1.896  | 1.412  | 5.163    |
> | BiMediX2 (8B)                      | 0.405  | 0.428  | 0.315  | 0.715  | 1.863    |
> | Lingshu (32B)                      | 0.678  | 0.751  | 1.955  | 1.105  | 4.489    |
> | MedGemma (27B)                     | 0.796  | 0.523  | 1.687  | 1.314  | 4.320    |
> | Mistral-Medium-3                   | 0.987  | 0.878  | 1.742  | 1.445  | 5.052    |
> | Claude-4-Sonnet                    | 0.914  | 0.787  | 1.869  | 1.476  | 5.046    |
> | Gemini-2.5-Pro                     | 0.905  | 0.860  | 1.710  | 1.605  | 5.080    |
> | Grok-4                             | 1.095  | 1.005  | 1.823  | 1.550  | 5.473    |
> | GPT-4o                             | 1.077  | 1.036  | 1.860  | 1.520  | 5.493    |
> | GPT-5                              | **1.220** | **1.145** | 1.850  | **1.630** | **5.845** |

---

> ### Author Response · Authors · 2025-11-23
> **Response to Reviewer wTsk (3/3)**
>
> > - W3: A more in-depth analysis of concrete failure modes would be helpful with some discussion on how to address the shortcomings of current models, highlighting potential future directions in model design or training improvements.
> > - Q3: What are the technical implications of the paper on the design and development of medical MLLMs?
>
> **A3:** We have expanded Appendix A.6 to provide a clearer synthesis of concrete failure modes and their technical implications. Our analysis identifies four recurring issues: limited understanding of modality-specific low-level attributes, difficulty with fine-grained distinctions such as mild degradations, limited quality-reasoning capability rooted in weak perceptual foundations, and diagnostic bias in medical-specialized models that conflate recognizability with quality. These findings point to actionable directions for improving medical MLLMs, including modality-aware quality training, fine-grained perceptual enhancement through degradation-focused pretraining, RL-based structured reasoning alignment using dedicated quality-reasoning objectives, and training strategies that explicitly decouple diagnostic reasoning from low-level quality assessment.

---

### Official Review · Reviewer_GiGe · 2025-10-30

**Soundness:** 2
**Presentation:** 2
**Contribution:** 2
**Rating:** 4
**Confidence:** 4

**Summary:**

This work proposes MedQ-Bench, a comprehensive benchmark that establishes a perception–reasoning paradigm for language-based evaluation of medical image quality with MLLMs. This work conducts a thorough evaluation of 14 state-of-the-art MLLMs, demonstrating that models exhibit preliminary but unstable perceptual and reasoning skills, with insufficient accuracy for reliable clinical use.

**Strengths:**

1. This work introduces a comprehensive benchmark with 2,600 perceptual queries and 708 reasoning assessments, establishing a perception–reasoning paradigm for language-based evaluation of medical image quality with MLLMs.
2. This work conducts a thorough evaluation of 14 state-of-the-art MLLMs. The evaluation results indicate that the models demonstrate preliminary yet unstable perceptual and reasoning skills.

**Weaknesses:**

1. As the main scope is about medical images, why are AI-generated medical images collected? In reality, only real medical images will be used. Maybe some AI-generated medical images will contain some strange artifacts, which never occur in real cases. Therefore, I do not understand why this benchmark includes these AI-generated medical images.

2. Line 68, SSIM is proposed in [R1]. The authors cite the wrong paper. Please do not simply search on Google Scholar. Make sure that every paper you cited is carefully checked.

[R1] Wang, Zhou, et al. "Image quality assessment: from error visibility to structural similarity." IEEE transactions on image processing 13.4 (2004): 600-612.

3. Now, the word "reasoning" has some specific meanings, i.e., reasoning-based models like OpenAI-o1 and DeepSeek-R1. It is better to rename this task to "interpretation". I know that some prior works use this word, but at that time, the word "reasoning" was not so specific.

4. Dataset annotation is a very important process in benchmark construction. I would like to suggest that the authors move some parts from the Appendix to the main paper.

5. I am very confused about Figure 2. In this figure, the authors use "Reasoning-IQA (Ours)". However, this work is actually a benchmark work, and this work does not train its own medical IQA model. What is the meaning of "Reasoning-IQA (Ours)"?

6. From Line 062 to Line 093, this part does not have any relationship to the main scope of this work. The main scope of this work is to assess the abilities of MLLMs in the medical IQA field. Why use such a large space to discuss score-based IQA? If this work discusses the weaknesses of score-based IQA, so what new medical IQA method does this work introduce? Actually, this work does not propose any new models. Overall, the introduction section is very chaotic, like a mixture of a method paper and a benchmark paper.

**Questions:**

1. In Figure 3 "Simulate Artifact", " Overall, the quality of this image is reject" should be "Overall, the quality of this image is rejected".
2. Some section names end without "." like "2.3.3 EVALUATION METRICS", but some with like "2.3.2 COMPARISON REASONING TASKS.".
3. In Line 765, the closing quotation mark ('' in latex) should be the opening quotation mark (`` in latex). The same in Line 812, Line 813, and Line 814.

---

> ### Author Response · Authors · 2025-11-25
> **Response to Reviewer GiGe (1/3)**
>
> Thank you very much for your constructive comments. We have carefully revised the paper in line with your suggestions. Below we provide our detailed responses, and we would be grateful if you could let us know whether our revisions adequately address your concerns.
>
> ---
> > W1: As the main scope is about medical images, why are AI-generated medical images collected? In reality, only real medical images will be used. Maybe some AI-generated medical images will contain some strange artifacts, which never occur in real cases. Therefore, I do not understand why this benchmark includes these AI-generated medical images.
>
> **A1:** We understand the reviewer’s concern regarding the inclusion of AI-generated medical images. In practice, AI-processed images are already becoming part of real clinical and computational workflows, including generative-model–based MRI accelerated reconstruction [3], cross-modal synthesis for missing-sequence completion [4], and generative augmentation used in training diagnostic models [1,2]. These images influence both clinical interpretation and downstream model behavior, so evaluating an MLLM’s ability to reason about the quality of AI-derived images is clinically relevant. Moreover, AI-generated images exhibit degradation patterns that are difficult to collect through real acquisitions or physics-based simulations. Excluding them would therefore leave a systematic gap in assessing whether MLLMs can identify generative algorithm-induced quality risks.
>
> References:
>
> [1] Chen et al., "Towards generalizable tumor synthesis," CVPR 2024.
>
> [2] Wu et al., "Large-Scale Generative Tumor Synthesis in Computed Tomography Images for Improving Tumor Recognition," Nature Communications, 2025.
>
> [3] Chung et al., "Decomposed Diffusion Sampler for Accelerating Large-Scale Inverse Problems," ICLR 2024.
>
> [4] Dalmaz et al., "ResViT: Residual vision transformers for multimodal medical image synthesis," TMI 2022.

---

> ### Author Response · Authors · 2025-11-25
> **Response to Reviewer GiGe (2/3)**
>
> > - W2: Line 68, SSIM is proposed in [R1]. The authors cite the wrong paper. Please do not simply search on Google Scholar. Make sure that every paper you cited is carefully checked.
> > - Q1: In Figure 3 "Simulate Artifact", " Overall, the quality of this image is reject" should be "Overall, the quality of this image is rejected".
> > - Q2: Some section names end without "." like "2.3.3 EVALUATION METRICS", but some with like "2.3.2 COMPARISON REASONING TASKS.".
> > - Q3: In Line 765, the closing quotation mark ('' in latex) should be the opening quotation mark (`` in latex). The same in Line 812, Line 813, and Line 814.
>
> **A2:** We have corrected and verified all these similar problems throughout the manuscript. Regarding the phrase “the quality of this image is reject,” we note that “reject” is a predefined categorical label in our benchmark (parallel to “good” and “usable”), not a verb requiring the “-ed” form.
>
> ---
> > W3: Now, the word "reasoning" has some specific meanings, i.e., reasoning-based models like OpenAI-o1 and DeepSeek-R1. It is better to rename this task to "interpretation". I know that some prior works use this word, but at that time, the word "reasoning" was not so specific.
>
> **A3:** We appreciate and agree with the reviewer’s suggestion. The term “interpretation” offers a clearer and less overloaded description of the task, especially given recent developments in LLM terminology. As several places in the manuscript require careful proofreading for consistency, we will adopt “interpretation” as the unified term and update the terminology throughout the camera-ready version if the paper is accepted.

---

> ### Author Response · Authors · 2025-11-25
> **Response to Reviewer GiGe (3/3)**
>
> > W4: Dataset annotation is a very important process in benchmark construction. I would like to suggest that the authors move some parts from the Appendix to the main paper.
>
> **A4:** Due to space constraints, the detailed annotation protocols were originally placed in Appendix A.1. In response to the reviewer’s suggestion, we have revised the paper and put some parts in Section 2.4.
>
> ---
> > W5: I am very confused about Figure 2. In this figure, the authors use "Reasoning-IQA (Ours)". However, this work is actually a benchmark work, and this work does not train its own medical IQA model. What is the meaning of "Reasoning-IQA (Ours)"?
>
> **A5:** The label “Reasoning-IQA (Ours)” in Figure 2 may unintentionally suggest that we trained a new IQA model, which is not the case; it refers only to our evaluation paradigm, reasoning-based quality assessment under the MedQ-Bench protocol, rather than to a learned model. To avoid confusion, we have updated Figure 2 with clearer labels and a revised caption in the manuscript.
>
> ---
> > W6:
> From Line 062 to Line 093, this part does not have any relationship to the main scope of this work. The main scope of this work is to assess the abilities of MLLMs in the medical IQA field. Why use such a large space to discuss score-based IQA? If this work discusses the weaknesses of score-based IQA, so what new medical IQA method does this work introduce? Actually, this work does not propose any new models. Overall, the introduction section is very chaotic, like a mixture of a method paper and a benchmark paper.
>
> **A6:** We appreciate the reviewer’s structural advice regarding the discussion of score-based IQA in the introduction. Our intention is to highlight that existing score-based paradigms suffer from limited generalization and lack interpretable reasoning; while MLLMs could address these issues, there is currently no systematic way to evaluate their capabilities. MedQ-Bench is designed to fill this gap. To improve clarity, we have revised the introduction in the manuscript.

---

> > ### Comment · Reviewer_GiGe · 2025-11-27
> > **Response**
> >
> > My concerns about AIGC images remain. The evaluation of AIGC medical images and real medical images is totally different. The evaluation of AIGC medical images should focus on whether the generated image is real or fake, while the evaluation of real medical images should focus on the image quality like noise, brightness, blur.
> >
> > Moreover, considering the chaotic writing, careless citation, confusing presentation, I keep my original borderline reject rating.

---

> > > ### Author Response · Authors · 2025-11-27
> > >
> > > Dear Reviewer GiGe,
> > >
> > > We sincerely thank you for your feedback. We would like to emphasize again that including AIGC images is not optional but **necessary** for ensuring the completeness, coverage, and practical relevance of MedQ-Bench. Modern medical imaging pipelines already rely on reconstruction, enhancement, and synthesis models, and these systems inevitably introduce low-level degradations. Excluding AIGC data would create a systematic blind spot and prevent the benchmark from evaluating whether MLLMs can detect the quality issues that arise specifically from AI-based reconstruction and synthesis.
> > >
> > > We also clarify that determining whether an image is real or fake belongs to authenticity and safety evaluation, which is outside the intended scope of a quality-focused benchmark. MedQ-Bench is designed to isolate low-level quality assessment rather than clinical semantic correctness or authenticity detection.
> > >
> > > Regarding the reviewer’s comments on writing, citation accuracy, and clarity, we have now thoroughly verified all references, reorganized the introduction, and substantially improved the overall readability of the manuscript.

---

### Official Review · Reviewer_PPyC · 2025-10-31

**Soundness:** 3
**Presentation:** 2
**Contribution:** 3
**Rating:** 4
**Confidence:** 3

**Summary:**

The paper introduces MedQ-Bench, a novel benchmark for evaluating the medical image quality assessment (IQA) capabilities of multimodal large language models (MLLMs). MedQ-Bench establishes a perception-reasoning paradigm with two complementary tasks: MedQ-Perception (probing low-level visual attributes) and MedQ-Reasoning (encompassing no-reference and comparison reasoning). The benchmark covers 5 imaging modalities and over 40 quality attributes, comprising 2,600 perceptual queries and 708 reasoning assessments. The authors propose a multi-dimensional judging protocol for reasoning tasks and validate it through human-AI alignment. Evaluations of 14 MLLMs reveal preliminary but unstable perceptual and reasoning skills, highlighting the need for targeted optimization in medical IQA.

**Strengths:**

1) Addresses a critical need for evaluating MLLMs in the safety-critical domain of medical IQA.
2) MedQ-Bench is a comprehensive and well-designed benchmark, covering diverse modalities, quality attributes, and image sources.
3) The multi-dimensional judging protocol and human-AI alignment validation enhance the reliability and validity of the evaluation.

**Weaknesses:**

- The reasoning tasks inherently involve subjectivity, and the paper could provide more detail on how the annotation process minimizes bias and ensures reliability.
- What are the most common types of errors made by MLLMs on MedQ-Bench? Are there specific IQA tasks or degradation types that are particularly challenging?
- What are some potential applications of MedQ-Bench beyond model evaluation? Could it be used to guide the development of new IQA algorithms or assist clinicians in quality control?
Computational Cost: What is the computational cost of using MedQ-Bench for evaluation? How long does it take to evaluate a single model on the benchmark?
- Report inter-annotator agreement metrics (e.g., Cohen's kappa) for the reasoning tasks.
Discuss how disagreements between annotators were resolved.

**Questions:**

- Some models (e.g., Grok-4, Qwen2.5-VL) performed better on fine-grained tasks than coarse-grained ones (Figure 6). This is counter-intuitive. Do you have a hypothesis for this phenomenon?
- How did you mitigate the risk that using GPT-4o for question generation would unfairly favor GPT-family models in the evaluation? Have you analyzed whether there is a "style bias" in the generated questions?
-  While the human-AI alignment is strong on the 200 sampled cases, how can you be sure this alignment holds for the full diversity of outputs from all 14 models, especially the very poor or nonsensical responses from weaker models?

**Details Of Ethics Concerns:**

N.A

---

> ### Author Response · Authors · 2025-11-23
> **Response to Reviewer PPyC (1/4)**
>
> Thank you for reviewing our paper and for your valuable feedback. Below, we address your concerns point by point, and we’ve revised our paper according to your suggestions. We would appreciate it if you could let us know whether your concerns are addressed by our response.
>
> ---
> > - W1: The reasoning tasks inherently involve subjectivity, and the paper could provide more detail on how the annotation process minimizes bias and ensures reliability.
> > - W4: Report inter-annotator agreement metrics (e.g., Cohen's kappa) for the reasoning tasks. Discuss how disagreements between annotators were resolved.
>
> **A1:** To minimize bias and ensure annotation reliability, we used a structured multi-round expert protocol. Three medical imaging specialists first annotated each case independently following a standardized four-step procedure—visual inspection, modality-specific degradation assessment, severity grading, and structured reasoning—supported by interfaces that constrain arbitrary variation. Annotators then cross-checked labels and resolved divergent cases through focused discussion using explicit visual and clinical evidence; when consensus could not be reached, the final label was assigned by majority vote. To assess reliability, we computed Cohen’s $\kappa$ between each annotator and the majority-vote reference label: $\kappa$ = 0.873 for perception tasks, and the $\kappa$ values for reasoning tasks are reported in Table T1. These results indicate substantial to near-perfect agreement. Details are now included in Sections A.1 and A.5.3.
>
> **Table T1.** Inter-annotator agreement (Cohen’s $\kappa$) for reasoning tasks.
>
> | Metric    | Completeness | Preciseness | Consistency | Quality Classification |
> |-----------|--------------|-------------|-------------|-------------------------|
> | $\kappa$ | 0.843         | 0.817        | 0.856        | 0.884                    |

---

> ### Author Response · Authors · 2025-11-23
> **Response to Reviewer PPyC (2/4)**
>
> > - W2: What are the most common types of errors made by MLLMs on MedQ-Bench? Are there specific IQA tasks or degradation types that are particularly challenging?
>
> **A2:** We analyzed GPT-5’s failure patterns to identify which inputs contribute most to its errors on MedQ-Bench, as shown in Table T2. At the artifact level, the most challenging categories are Air Bubbles, Specularity, and Blood-related artifacts, with error rates of 95.23%, 90.48%, and 84.62%, respectively. We have revised the paper and put these details in Appendix A.5.2.
>
> **Table T2:** GPT-5 Error Rates Across the Most Challenging Modalities.
> | Modality                    | Error Rate |
> |-----------------------------|------------|
> | Endoscopy                   | 39.11%     |
> | Histopathology | 34.57%     |
> | Fundus                      | 29.91%     |
> | CT                          | 25.76%     |
> | MRI                         | 24.10%     |

---

> ### Author Response · Authors · 2025-11-23
> **Response to Reviewer PPyC (3/4)**
>
> > - W3: What are some potential applications of MedQ-Bench beyond model evaluation? Could it be used to guide the development of new IQA algorithms or assist clinicians in quality control? Computational Cost: What is the computational cost of using MedQ-Bench for evaluation? How long does it take to evaluate a single model on the benchmark?
>
> **A3:** Beyond model evaluation, MedQ-Bench offers broader methodological and practical value. It provides a standardized platform for analyzing the strengths and limitations of IQA and quality-reasoning algorithms, guiding the design of models that target specific perceptual or reasoning skills. Its three-tier quality system aligns with real clinical decision workflows and can support the development of automatic quality-control modules in radiology practice. MedQ-Bench also enables the construction of targeted training data for improving low-level visual perception in medical MLLMs and serves as a structured resource for educating technologists and radiology trainees in systematic quality assessment.
> Regarding computational cost, API-based models and locally deployed models can complete the full benchmark within short and predictable runtime budgets, and the associated API expenses remain low. Detailed runtime and cost characteristics for different model types are summarized in Table T3.
>
> **Table T3:** Summary of runtime and API cost for single-model evaluation on MedQ-Bench.
> | Evaluation Setting                             | Time Cost            | API Monetary Cost |
> |-------------------------------------------------|-----------------------|---------------------|
> | API-evaluated model (GPT-4o, Claude, Gemini, etc.) | 4–8 minutes     | $5–7               |
> | Small deployed model (7B–8B)                | 20–60 minutes         | $0.93              |
> | Medium deployed model (32B–38B)             | 1.4–4.3 hours         | $0.93              |
> | Large deployed model (72B+, multi-GPU)      | 4–9 hours             | $0.93              |

---

> ### Author Response · Authors · 2025-11-23
> **Response to Reviewer PPyC (4/4)**
>
> > - Q1: Some models (e.g., Grok-4, Qwen2.5-VL) performed better on fine-grained tasks than coarse-grained ones (Figure 6). This is counter-intuitive. Do you have a hypothesis for this phenomenon?
>
> **A4:** As suggested by Fig. 6, a plausible explanation is that models such as Grok-4 and Qwen2.5-VL tend to engage in more careful, attribute-level inspection when differences between two images are subtle, whereas in coarse-grained settings they rely on more global, heuristic judgments that can lead to overconfident mistakes. This behavior may also stem from their pretraining distribution: models exposed to a wider range of mild degradations than extreme ones are naturally better calibrated for fine-grained comparisons.
>
> ---
> > Q2: How did you mitigate the risk that using GPT-4o for question generation would unfairly favor GPT-family models in the evaluation? Have you analyzed whether there is a "style bias" in the generated questions?
>
> **A5:** We agree that question-generation bias is an important fairness concern, and this issue was explicitly considered during the design of MedQ-Bench. In our benchmark, GPT-4o did not create the question templates; all core question structures were derived entirely from expert-designed seed templates that define the intended semantics, difficulty, and reasoning scope. GPT-4o was used only to expand these seeds under strict semantic and structural constraints. Every expanded question was then reviewed and revised by three independent medical imaging experts to ensure that wording, phrasing, and difficulty remained consistent with the expert templates and did not drift toward any model-specific stylistic patterns.
>
> ---
> > Q3: While the human-AI alignment is strong on the 200 sampled cases, how can you be sure this alignment holds for the full diversity of outputs from all 14 models, especially the very poor or nonsensical responses from weaker models?
>
> **A6:** To assess whether human–AI alignment remains stable across quality levels, our 200 sampled outputs were partitioned into three tiers by human-score quantiles, and we examined the lowest third. As shown in Table T4, the weighted $\kappa$ values show some decrease in this subset, which contains more low-quality or failed predictions, but the alignment does not substantially degrade even under these challenging cases. We have already revised the paper and put these details in Appendix A.5.4
>
> **Table T4:** Human–AI Alignment in the Lowest-Quality Tier
> | Metric        | Completeness | Preciseness | Consistency | Quality Accuracy |
> |---------------|--------------|-------------|-------------|------------------|
> | $\kappa$       | 0.702        | 0.791       | 0.755       | 0.952            |

---

### Official Review · Reviewer_q3Dd · 2025-10-31

**Soundness:** 2
**Presentation:** 2
**Contribution:** 2
**Rating:** 4
**Confidence:** 5

**Summary:**

The paper proposes MedQ-Bench, which it describes as “the first comprehensive benchmark” for evaluating medical image quality assessment (IQA) abilities of multimodal large language models (MLLMs). The benchmark is organized around a “perception–reasoning paradigm” with two main tasks: (1) MedQ-Perception, which uses curated visual questions to test perceptual understanding of low-level quality attributes, and (2) MedQ-Reasoning, which asks models to generate free-form analyses of quality issues, including comparative reasoning across image pairs. The dataset spans 5 imaging modalities, covers 40+ quality attributes, and claims 2,600 perception queries and 708 reasoning assessments across 3,308 samples.

**Strengths:**

The topic (MLLMs for medical IQA) is timely and important. The benchmark could become useful, and the clinical framing is compelling. The data is comprehensive, and the evaluation is extensive.

**Weaknesses:**

1. Novelty is overstated. The benchmark looks like a domain-specific adaptation of prior “quality reasoning” benchmarks. In Related Work, the paper itself cites several highly similar efforts in non-medical domains, e.g., Q-Bench. These explicitly aim to (i) probe perception of low-level degradations, (ii) elicit natural-language quality rationales, and (iii) score reasoning-like explanations. This work is more of an adaptation of an existing template

2. The benchmark tasks are (a) Yes/No / What / How multiple-choice perceptual questions, and (b) free-form textual reasoning that ends in a 3-level “good/usable/reject” judgment, which is said to approximate clinical acceptability. However, there is no evidence that performance on these tasks correlates with actual downstream diagnostic safety, e.g. sensitivity/specificity of pathology detection by radiologists when given “usable” vs “reject” images.

3. There is no reader study showing that if an MLLM calls an image “usable,” a clinician would skip reacquisition and would not miss a lesion. This claim and setting are just too intuitive. The paired comparison task is mostly about visual preference (“which image is higher diagnostic quality”), not about safety-critical failure modes.

4. To score the free-form reasoning outputs, the authors introduce a GPT-4o-based measure suite. And they claim that recent studies have demonstrated GPT-4o to be a reliable evaluation tool and perform human–AI alignment validation on 200 sampled cases. You use GPT-4o as both (i) the evaluation judge for all models and (ii) a system under evaluation in Table 1. LLMs-as-judges prefer to vote the one with a similar corpus distribution to itself.  This raises fairness concerns: is GPT-4o systematically more lenient to GPT-4o-like phrasing styles?

5. For perception, results are reported as aggregate accuracies for Yes-or-No, What, and How, plus an “Overall” number per model. This is overly simplistic and does not regard the uniqueness of the medical scenario. No significance testing or CIs are reported. For example, is GPT-5’s 68.97% actually significantly better than GPT-4o’s 64.79%?

6. The pipeline relied on GPT-4o for question expansion. That means the benchmark may encode GPT-4o’s linguistic priors, which in turn advantages GPT-4o (or the entire GPT-family if stylistically similar) during evaluation.

7. The paper does not give a quantitative definition of “fine-grained difference.” How was this threshold established? By MOS? By PSNR/SSIM gap? By the qualitative agreement of three experts?

8. Our primary contributions are as follows: repeats twice.

9. LVLMs and MLLMs are used alternately. Please be consistent on terminology.

10. Section 2.3.1 says reasoning tasks avoid high-level diagnostic interpretation and focus on low-level technical quality factors. But examples in Figure 3 include modality and anatomical region identification. Why? Please clarify the line between “quality reasoning” and “diagnosis.”

**Questions:**

Please see the section above.

---

> ### Author Response · Authors · 2025-11-24
> **Response to Reviewer q3Dd (1/6)**
>
> Thank you for carefully reviewing our paper and for the constructive feedback. We address each concern point by point below and have revised the manuscript accordingly. We would greatly appreciate your confirmation on whether our responses satisfactorily resolve the issues you raised.
>
> ---
> > W1: Novelty is overstated. The benchmark looks like a domain-specific adaptation of prior “quality reasoning” benchmarks. In Related Work, the paper itself cites several highly similar efforts in non-medical domains, e.g., Q-Bench. These explicitly aim to (i) probe perception of low-level degradations, (ii) elicit natural-language quality rationales, and (iii) score reasoning-like explanations. This work is more of an adaptation of an existing template
>
> **A1:** We appreciate the reviewer’s concern and understand why MedQ-Bench may initially appear to be a domain-specific adaptation of prior quality-reasoning datasets. However, our goal is not simply porting. *We aim to address gaps that are unique to clinical imaging and that existing benchmarks cannot cover.* Medical image quality is determined by acquisition physics, modality protocols, and clinically meaningful artifact types that do not exist in natural-image IQA. To capture these, MedQ-Bench introduces three domain-specific components: (1) quality attributes grounded in medical acquisition and degradation mechanisms rather than generic distortions; (2) evaluation design aligned with clinical practice, including multidimensional acceptability labels, paired comparisons that reflect clinician decision tradeoffs, and reasoning prompts focused on technical quality factors; and (3) expert grounding through multi-round annotation and structured consensus to ensure clinical validity rather than template adaptation. Taken together, these elements show that MedQ-Bench is not a straightforward extension but a benchmark purpose-built to evaluate medical-specific quality reasoning and robustness that prior work does not address, and we believe it fills a gap that natural-image benchmarks such as Q-Bench cannot cover.

---

> ### Author Response · Authors · 2025-11-24
> **Response to Reviewer q3Dd (2/6)**
>
> > - W2: The benchmark tasks are (a) Yes/No / What / How multiple-choice perceptual questions, and (b) free-form textual reasoning that ends in a 3-level “good/usable/reject” judgment, which is said to approximate clinical acceptability. However, there is no evidence that performance on these tasks correlates with actual downstream diagnostic safety, e.g. sensitivity/specificity of pathology detection by radiologists when given “usable” vs “reject” images.
> > - W3: There is no reader study showing that if an MLLM calls an image “usable,” a clinician would skip reacquisition and would not miss a lesion. This claim and setting are just too intuitive. The paired comparison task is mostly about visual preference (“which image is higher diagnostic quality”), not about safety-critical failure modes.
>
> **A2:** MedQ-Bench is designed to evaluate foundational quality-perception and reasoning abilities rather than to model the full clinical workflow from quality assessment to diagnostic decision-making. Our tasks target low-level quality attributes that universally influence diagnostic interpretability. All labels were obtained through expert consensus, ensuring that the quality judgments themselves are clinically grounded. That said, demonstrating a direct correlation between MLLM quality predictions and downstream diagnostic safety would require dedicated reader studies. Such studies must evaluate how radiologists interpret pathology under controlled quality conditions and rely on carefully curated diagnostic ground truth. They also require substantial clinical resources and logistics. For these reasons, this type of clinical validation is beyond the scope of a low-level benchmark paper. We acknowledge this limitation. We have already revised the paper and discussed it in Appendix A.6.

---

> > ### Author Response · Authors · 2025-11-24
> > **Response to Reviewer q3Dd (5/6)**
> >
> > > - W6: The pipeline relied on GPT-4o for question expansion. That means the benchmark may encode GPT-4o’s linguistic priors, which in turn advantages GPT-4o (or the entire GPT-family if stylistically similar) during evaluation.
> >
> > **A5:**  We agree that question-generation bias is an important fairness concern, and this issue was explicitly considered during the design of MedQ-Bench. In our benchmark, GPT-4o did not create the question templates; all core question structures were derived entirely from expert-designed seed templates that define the intended semantics, difficulty, and reasoning scope. GPT-4o was used only to expand these seeds under strict semantic and structural constraints. Every expanded question was then reviewed and revised by three independent medical imaging experts to ensure that wording, phrasing, and difficulty remained consistent with the expert templates and did not drift toward any model-specific stylistic patterns.
> >
> > ---
> > > - W7: The paper does not give a quantitative definition of “fine-grained difference.” How was this threshold established? By MOS? By PSNR/SSIM gap? By the qualitative agreement of three experts?
> >
> > **A6:** The fine-grained versus coarse-grained distinction was defined through a structured expert review process. Three board-certified radiologists independently reviewed all image pairs and categorized them based on how difficult the quality difference is to perceive: coarse-grained pairs exhibit clear, immediately noticeable degradations, whereas fine-grained pairs involve subtle differences that require deliberate inspection. After independent labeling, disagreements were resolved through discussion and consensus, ensuring a stable expert standard. We have revised the paper and put these details in Appendix A.1.

---

> ### Author Response · Authors · 2025-11-24
> **Response to Reviewer q3Dd (3/6)**
>
> > - W4: To score the free-form reasoning outputs, the authors introduce a GPT-4o-based measure suite. And they claim that recent studies have demonstrated GPT-4o to be a reliable evaluation tool and perform human–AI alignment validation on 200 sampled cases. You use GPT-4o as both (i) the evaluation judge for all models and (ii) a system under evaluation in Table 1. LLMs-as-judges prefer to vote the one with a similar corpus distribution to itself. This raises fairness concerns: is GPT-4o systematically more lenient to GPT-4o-like phrasing styles?
>
> **A3:** To directly assess the fairness of using GPT-4o as a judge, we conducted a multi-judge validation study with three independent LLM judges (GPT-4o, Claude-4-Sonnet, and Gemini-2.5-Pro), as shown in Table T1. The three judges produce highly consistent scores, with inter-judge differences typically below 0.05 and the Intraclass Correlation Coefficient (ICC) values exceeding 0.95. Importantly, GPT-4o does not assign higher scores to OpenAI models than the other judges, indicating no self-preference. We have added these experiments and analyses in Appendix A.5.4.
>
> **Table T1:** Multi-judge validation results showing overall reasoning scores from three independent LLM judges and inter-judge consistency metrics. MAD (Claude) and MAD (Gemini) represent the mean absolute deviation (MAD) between GPT-4o scores and Claude/Gemini scores, respectively.
> | Model                         | GPT-4o | Claude | Gemini | ICC   | MAD (Claude) | MAD (Gemini) |
> |------------------------------|-------|--------|--------|-------|--------------|--------------|
> | Qwen2.5-VL-Instruct (7B)     | 4.367 | 4.376  | 4.371  | 0.984 | 0.009        | 0.004        |
> | Qwen2.5-VL-Instruct (32B)    | 5.272 | 5.294  | 5.308  | 0.992 | 0.022        | 0.036        |
> | InternVL3 (8B)               | 4.983 | 4.995  | 4.968  | 0.986 | 0.012        | 0.015        |
> | InternVL3 (38B)              | 4.965 | 4.977  | 4.977  | 0.990 | 0.012        | 0.012        |
> | Qwen2.5-VL-Instruct (72B)    | 4.982 | 4.932  | 4.959  | 0.988 | 0.050        | 0.023        |
> | BiMediX2 (8B)                | 1.721 | 1.692  | 1.719  | 0.954 | 0.029        | 0.002        |
> | Lingshu (32B)                | 4.312 | 4.344  | 4.317  | 0.987 | 0.032        | 0.005        |
> | MedGemma (27B)               | 4.054 | 4.036  | 4.036  | 0.978 | 0.018        | 0.018        |
> | Mistral-Medium-3             | 4.557 | 4.529  | 4.520  | 0.988 | 0.028        | 0.037        |
> | Claude-4-Sonnet              | 4.529 | 4.529  | 4.570  | 0.981 | 0.000        | 0.041        |
> | Gemini-2.5-Pro               | 5.018 | 5.014  | 5.005  | 0.985 | 0.004        | 0.013        |
> | GPT-4o                       | 5.321 | 5.335  | 5.321  | 0.989 | 0.014        | 0.000        |
> | GPT-5                        | 5.679 | 5.679  | 5.652  | 0.987 | 0.000        | 0.027        |

---

> ### Author Response · Authors · 2025-11-24
> **Response to Reviewer q3Dd (4/6)**
>
> > - W5: For perception, results are reported as aggregate accuracies for Yes-or-No, What, and How, plus an “Overall” number per model. This is overly simplistic and does not regard the uniqueness of the medical scenario. No significance testing or CIs are reported. For example, is GPT-5’s 68.97% actually significantly better than GPT-4o’s 64.79%?
>
> **A4:** To provide statistically rigorous comparisons, we now conduct paired t-tests for all test models, as shown in Table T2. We have revised the paper and put these details in Section 3.1.
>
> **Table T2:** Statistical significance between GPT-5 and the other models is assessed using paired t-tests (p < 0.05), with significant differences marked by asterisks in the Perception Overall results on the test set.
> | Model                       | Overall (%) | p-value  |
> |-----------------------------|-------------|----------|
> | Qwen2.5-VL-Instruct (7B)    | 54.71*      | <10e-6   |
> | Qwen2.5-VL-Instruct (32B)   | 59.31*      | 3.00e-10 |
> | InternVL3 (8B)              | 60.08*      | 2.30e-9  |
> | InternVL3 (38B)             | 61.00*      | 2.01e-7  |
> | Qwen2.5-VL-Instruct (72B)   | 63.14*      | 7.82e-5  |
> | BiMediX2 (8B)               | 35.10*      | <10e-6   |
> | Lingshu (32B)               | 50.88*      | <10e-6   |
> | MedGemma (27B)              | 57.16*      | <10e-6   |
> | Mistral-Medium-3            | 57.70*      | <10e-6   |
> | Claude-4-Sonnet             | 60.23*      | 1.63e-8  |
> | Gemini-2.5-Pro              | 61.88*      | 2.30e-9  |
> | Grok-4                      | 63.14*      | 2.16e-5  |
> | GPT-4o                      | 64.79*      | 1.44e-3  |
> | GPT-5                       | 68.97       | –        |

---

> ### Author Response · Authors · 2025-11-24
> **Response to Reviewer q3Dd (6/6)**
>
> > - W8: Our primary contributions are as follows: repeats twice.
> > - W9: LVLMs and MLLMs are used alternately. Please be consistent on terminology.
>
> **A7:**  Both issues in W8 and W9 have been resolved in the revised manuscript.
>
> ---
> > - W10: Section 2.3.1 says reasoning tasks avoid high-level diagnostic interpretation and focus on low-level technical quality factors. But examples in Figure 3 include modality and anatomical region identification. Why? Please clarify the line between “quality reasoning” and “diagnosis.”
>
> **A8:**  We understand the reviewer’s concern about the boundary between quality reasoning and diagnosis. The modality and anatomical-region identification in Figure 3 serve only as contextual cues that support technical quality assessment.  Knowing the modality allows the model to apply the correct modality-specific criteria, and knowing the anatomical region helps locate and describe where degradations occur. This contextual identification is categorically different from diagnostic reasoning, which involves interpreting disease features or making clinical judgments. Our benchmark focuses on low-level quality reasoning such as artifact identification, severity assessment, and visibility, without requiring any form of high-level clinical interpretation.

---

> > ### Comment · Reviewer_q3Dd · 2025-11-28
> > **Nice response!**
> >
> > Thanks for the detailed response from the authors. I think most of the concerns are well addressed. I tend to raise my score to 6 based on the current rebuttal, and I would raise it to 8 if the sixth point of the comment could be further discussed in the paper. But I think the score revision seems to be banned by the system currently. This comment can represent my updated rating if the technical issue cannot be further settled. Thanks.

---

> > > ### Author Response · Authors · 2025-11-28
> > >
> > > Dear Reviewer q3Dd,
> > >
> > > We sincerely thank you for your detailed and insightful feedback and for your willingness to raise the score to 8. We are glad that our response addressed your concerns. We have now incorporated an expanded discussion of W6 into Section 2.4 and Appendix A.1 of the manuscript. Thank you again for your valuable comments, which have helped us further improve the paper.

---

### Author Response · Authors · 2025-12-01
**Review and Reviewer-Author Discussion Summary (1/2)**

Dear PCs, SACs, ACs, and Reviewers,

Thank you very much for your valuable contributions to our work. To assist the newly assigned AC and help reduce their workload, we provide below a summary of the key points from the reviews and the reviewer-author discussions.

**Strength.** Overall, we are grateful that the reviewers gave this paper a positive evaluation in the initial reviews. Specifically:

**1. The benchmark addresses a critical and timely need in medical AI, establishing a comprehensive evaluation framework for MLLMs in medical image quality assessment.** All four reviewers explicitly recognized this contribution (q3Dd, PPyC, GiGe, wTsk).

**2. MedQ-Bench is comprehensive and well-designed, covering diverse modalities, quality attributes, and image sources with 2,600 perceptual queries and 708 reasoning assessments.** Three reviewers highlighted this strength (q3Dd, PPyC, wTsk).

**3. The multi-dimensional judging protocol and rigorous human-AI alignment validation enhance the reliability and validity of the evaluation.** Three reviewers explicitly praised this aspect (PPyC, wTsk, q3Dd).

**4. Questions are tailored to specific medical imaging modalities and reflect domain-specific quality attributes.** Two reviewers recognized this domain specificity (wTsk, PPyC).

**5. The evaluation of 14 state-of-the-art MLLMs is thorough and reveals important insights about current model limitations.** All reviewers acknowledged the extensive evaluation.

**6. The presentation is clear and well-motivated, with effective visualizations.** Two reviewers explicitly highlighted this (wTsk, PPyC).

---

> ### Author Response · Authors · 2025-12-01
> **Review and Reviewer-Author Discussion Summary (2/2)**
>
> **Concerns and Our Addressing**
>
> During the discussion period, we actively addressed the reviewers' concerns through substantial revisions and additional experiments, which were well-received by the reviewers. Specifically:
>
> **1. Evaluation Methodology and Fairness Concerns**
>
> Reviewers raised important questions about evaluation fairness and statistical rigor (q3Dd: W4-W6; PPyC: W1, W4; wTsk: W2).
>
> **Concern:** Using GPT-4o as both evaluation judge and evaluated system may introduce bias; lack of statistical significance testing; annotation reliability needs validation; subjective evaluation dimensions require calibration.
>
> **Our Addressing:** We conducted comprehensive validation experiments. Multi-judge validation (Response to q3Dd, **Table T1**) shows GPT-4o does not favor OpenAI models, eliminating bias concerns. Statistical significance testing (Response to q3Dd, **Table T2**) confirms all performance differences are significant (p < 0.05). Inter-annotator agreement (Response to PPyC, **Table T1**) demonstrates substantial agreement. Few-shot calibration (Response to wTsk, **Table T1**) shows consistent improvements. Human-AI alignment (Response to PPyC, **Table T4**) remains strong even for low-quality outputs. All three reviewers acknowledged these additions, with Reviewer q3Dd stating willingness to raise score to 8.
>
> **2. Clinical Scope and Benchmark Design**
>
> Reviewers questioned aspects of benchmark scope and design choices (q3Dd: W1-W3; wTsk: W1; GiGe: W1).
>
> **Concern:** Is MedQ-Bench merely adapting existing natural-image benchmarks? Does it address medical-specific needs? What about task-specific context-dependence of image quality? Why include AI-generated images?
>
> **Our Addressing:** We clarified the medical-specific design rationale. The benchmark is purpose-built with three domain-specific components: (1) quality attributes grounded in medical acquisition physics and modality-specific degradation mechanisms, (2) evaluation design aligned with clinical practice, and (3) rigorous expert grounding through multi-round annotation. The benchmark focuses on foundational, task-agnostic quality perception using expert-aligned scoring rubrics, with context-dependent extension discussed as future work (Appendix A.6). AI-generated images are included because AI-processed images already appear in clinical workflows and introduce unique degradation patterns essential for comprehensive evaluation.
>
> **3. Analysis Depth and Practical Value**
>
> Reviewers requested deeper analysis and practical guidance (PPyC: W2-W3; wTsk: W3).
>
> **Concern:** What are common failure modes? What are computational costs? What are technical implications for improving medical MLLMs?
>
> **Our Addressing:** We provided comprehensive analysis through error pattern analysis (Response to PPyC, **Table T2**, Appendix A.5.2) identifying the most challenging modalities and artifacts, and computational cost analysis (Response to PPyC, **Table T3**) demonstrating practical feasibility. We expanded Appendix A.6 to synthesize four recurring failure patterns and identify actionable improvement directions including modality-aware quality training, fine-grained perceptual enhancement, and RL-based reasoning alignment. We also discussed practical applications beyond model evaluation.
>
> **4. Presentation and Technical Details**
>
> Minor presentation and technical issues were raised (GiGe: W2-W6; q3Dd: W7-W10).
>
> **Our Addressing:** We thoroughly revised the manuscript: corrected citation errors, fixed grammatical issues, unified terminology, moved annotation details to Section 2.4, updated Figure 2, restructured the introduction, and clarified definitions (Appendix A.1).
>
> **Recognition of Our Revisions from Reviewers**
>
> One reviewer explicitly acknowledged that most of the concerns are well addressed and stated willingness to raise the score to 8. We believe that we have also properly addressed the remaining three reviewers' concerns.
>
> ---
> Above, we have faithfully summarized all reviewer comments and our corresponding responses, hoping that this will assist the AC's work. We are deeply grateful to the reviewers, AC, SAC, and PC, for their dedicated effort and excellent work. Their insightful feedback has further strengthened our paper. The authors offer their sincere respect and appreciation to all involved!
>
> Sincerely,
> Authors

---

### Note · Program_Chairs · 2026-01-17
**Submission Desk Rejected by Program Chairs**

The following references in this submission do not refer to real documents and/or have major errors in bibliographic information:

 Qian Wang, Baiqiao Liu, Hongjian Wang, Jiaheng Li, Qingyu Chen, and Zhiyong Lu. Lingshu: A linguistically-enhanced multi-modal chinese medical large language model. arXiv preprint arXiv:2406.06489, 2024b